# Discrimination reveals reconstructability of multiplex networks from partial observations

Mincheng Wu[1], Jiming Chen[1✉], Shibo He[1✉], Youxian Sun[1], Shlomo Havlin[2] & Jianxi Gao [3,4✉]

An excellent method for predicting links in multiplex networks is reflected in its ability to reconstruct them accurately. Although link prediction methods perform well on estimating the existence probability of each potential link in monoplex networks by the set of partially observed links, we lack a mathematical tool to reconstruct the multiplex network from the observed aggregate topology and partially observed links in multiplex networks. Here, we fill this gap by developing a theoretical and computational framework that builds a probability space containing possible structures with a maximum likelihood estimation. Then, we discovered that the discrimination, an indicator quantifying differences between layers from an entropy perspective, determines the reconstructability, i.e., the accuracy of such reconstruction. This finding enables us to design the optimal strategy to allocate the set of observed links in different layers for promoting the optimal reconstruction of multiplex networks. Finally, the theoretical analyses are corroborated by empirical results from biological, social, engineered systems, and a large volume of synthetic networks.

[1] State Key Laboratory of Industrial Control Technology, Zhejiang University, Hangzhou 310027, China. [2] Department of Physics, Bar-Ilan University, Ramat-Gan 52900, Israel. [3] Department of Computer Science, Rensselaer Polytechnic Institute, Troy, NY 12180, USA. [4] Network Science and Technology Center, Rensselaer Polytechnic Institute, Troy, NY 12180, USA. ✉email: cjm@zju.edu.cn; s18he@zju.edu.cn; gaoj8@rpi.edu

Multiplex networks, composed of a collection of layers sharing the same set of nodes, can describe multiple types of interactions between nodes in different layers more precisely than the corresponding aggregate networks[1–6]. Neglecting the layers topology in the multiplex structure may lead to significant inaccurate predictions[7–10]. For example, a tiny fraction of node removal in one layer may cause cascading failures between layers and a catastrophic collapse of the entire multiplex network[11] (Fig. 1a). However, the corresponding aggregate network may be predicted to remain unscathed for the same set of node removal (Fig. 1b), which are corroborated by recent results that an interdependent network is significantly more vulnerable than its isolated layers and from the aggregate network against random failures[12–14]. Another example is the random walk process over a multiplex transportation network. The cost of switching between two layers determines the navigability of a multiplex transportation network[15]. The walker may need an extra cost to switch from one layer to other layers (Fig. 1c), while one can walk along with any link without extra cost within the aggregate network (Fig. 1d). Such an extra cost causes a lower coverage of the nodes that are visited by the walker in a multiplex transportation network. A further example is the dynamic of the spreading process in a temporal network, modeling rumor circulation in a social network or disease outbreak (e.g., COVID-19) in a susceptible population network[16–19]. The topology of a temporal network may change at each time (Fig. 1e), while it is regarded as static in the aggregate network (Fig. 1f), leading to a lower infected fraction in a temporal network compared to the aggregate one.

However, it is practically difficult to obtain data representing the detailed layers of a full multiplex topology accurately, since it is costly, time-consuming, and even impossible to measure all types of interactions and heterogeneity of nodes, especially in a large-scale complex system. On the contrary, the aggregate network of a multiplex network is relatively feasible, aggregating all interactions without distinguishing their specific types. Researchers, for example, can construct the entire connectome of Caenorhabditis elegans' neural system[20,21] (Supplementary Fig. 1), potentially offering a better understanding of brains' functionality. However, this information is of limited use, as discussed above, because of the unidentified multiplex layers topology, reflecting on the types of interactions (e.g., gap-junction or synapse) between any two connected neurons without extensive experiments[22]. Analogous cases widely appear in various aspects of life, including social networks[23] and transportation networks[24]. Since dynamical phenomena on a multiplex network significantly differ from those on its aggregate one, it pressingly promotes the need for effective tools that can leverage limited information (the aggregate network) to accurately and efficiently predict unobserved links in each layer.

Recent studies have attempted to predict links in multiplex networks based on the structures (e.g., multiplex links and multilayer neighborhoods) when partial observations of a subnetwork in each layer are available[25,26]. These methods are efficient in retrieving missing links in different layers; however, they ignore the useful information of the aggregate network. Building on the results of link prediction, a more exciting and vital issue in multiplex networks is developing an efficient reconstruction method based on knowledge of both the aggregate topology and partial observations of layers beyond link prediction. Recent research demonstrates that one can reconstruct a multiplex network with a given generating model (e.g., the stochastic block model)[27–29] or specific dynamics data (e.g., random walk or a spreading process) is available[24,30]. Furthermore, efforts have been devoted to estimating the number of multiplex layers from the knowledge of the aggregate network or a dynamic process[24,31]. However, to the best of our knowledge, it remains lacking a framework to reconstruct multiplex layer structures and to display the specific topology of each hidden layer, when the number of layers is known. Thus, the following fundamental questions remain open.

The same aggregate network can be generated by different combinations of single-layer networks, which is a combinatorial optimization problem. There exists an enormous number of possible mappings from the potential multiplex layers structures to the observed aggregate topology. Specifically, the probability space composed by potential multiplex structures has an exponential $((2^L - 1)^{|A^{\mathcal{O}}|})$ possibilities with the number of layers $L$ and the number of links in the aggregate network $|A^{\mathcal{O}}|$ (see Supplementary Fig. 2 for more details). Q1. Can one conceive a low-complexity framework to reconstruct multiplex layer structures, avoiding the enormous cost of ergodic methods[32]? Various characteristics of a multiplex structure affect the reconstructability. For example, the average degrees and overlap of edges in different layers has a different impact on the reconstruction accuracy. Q2. Is there an indicator that can quantify the fundamental relation between the reconstructability and diverse network characteristics? More known links yield higher reconstruction accuracy while they are more costly. Furthermore, there is a huge number of possibilities to allocate a limited budget in different layers, which is also a combinatorial optimization problem. Q3. Is there an optimal strategy to allocate a limited budget that will enable the highest reconstruction accuracy for various multiplex networks using the above indicator?

In this article, we propose a mathematical and computational framework that can reconstruct a multiplex network and predict its dynamic process on it. We found a discrimination indicator derived from information entropy, integrated by multiple network characteristics, that linearly determines the reconstruction accuracy of multiplex networks. This discovery enables us to design an optimal strategy to allocate a fixed budget for partial observations of the layers, promoting the optimal reconstruction of layers in multiplex networks for the considered network model. Empirical results based on nine real-world multiplex networks and several synthetic networks corroborate our analytical results (see Supplementary Table 1 for details of the real-world datasets). Therefore, our answers to the open questions above are "yes".

## Results
**Framework for reconstructing multiplex layer structures**. We will first introduce the notations by denoting the adjacency matrix of layer $\alpha$ in a multiplex network $\mathbf{M}$ by $M^\alpha$ ($\alpha = 1, 2, \cdots, L$), and $M_{ij}^\alpha = 1$ if there is an edge between nodes $i, j$ in layer $\alpha$ and vice versa ($i, j = 1, 2, \cdots, N$). It is hypothesized that each multiplex network $\mathbf{M}$ is generated by some process such that the probability of generating a multiplex network with adjacency matrices $M^\alpha$ is $\prod_\alpha P(M^\alpha | \theta)$, where $\theta$ represents the parameters of such a process. An aggregate mechanism is a mapping from a multiplex network $\mathbf{M}$ to a monoplex (single-layer) topology $A^{\mathcal{O}}$, where $A^{\mathcal{O}} \in \mathbb{R}^{N \times N}$ describes the adjacency matrix. In this article, for illustration, we describe the framework using multiplex networks aggregated by the OR mechanism, which is the most common case ranging from biological networks to social networks (see Supplementary Note 1 for other aggregate mechanisms). Then, we have

$$A_{ij}^{\mathcal{O}} = 1 - \prod_{\alpha=1}^{L}(1 - M_{ij}^\alpha). \tag{1}$$

Partial observation $\Gamma$ indicates a set that contains the observed edges in the multiplex network, where $\Gamma = \{(i, j, \alpha) | A_{ij}^{\mathcal{O}} = 1, \text{ and}$

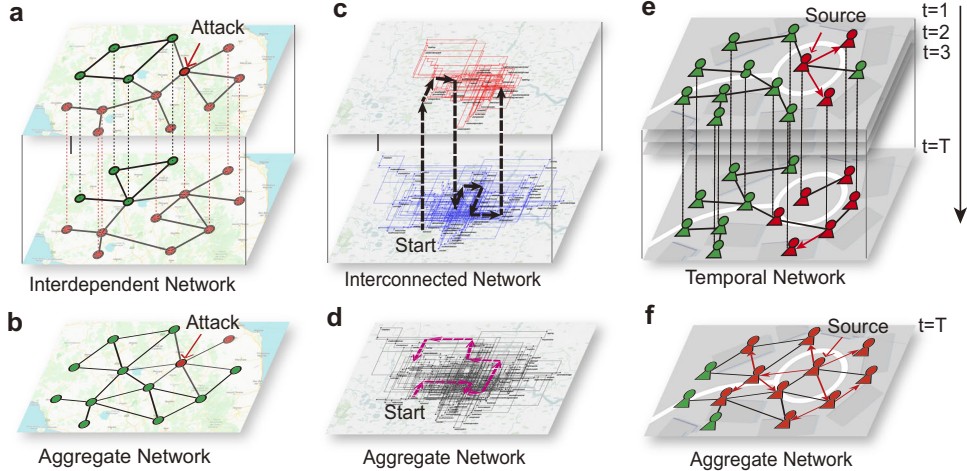

**Fig. 1 The different consequences for dynamics occurring on a multiplex network and the corresponding aggregate monoplex network. a** Once a node fails due to attack, the cascade of failures will result in a catastrophic removal of nodes (red nodes in the network) in an interdependent multiplex network. **b** Attacking the same node in the aggregate network will not trigger such a catastrophic outcome (only another one node disconnects from the giant connected component). **c** A random walk process in a multiplex transportation network, where the black dashed line indicates a possible path of the random walk process and the arrows indicate the direction of the random walk. **d** A random walk process in the corresponding aggregate network, where the red dashed line indicates a possible path of the random walk process. **e** In a temporal contact network, the topology changes in time and, as a consequence, an infection (red individuals) spreads slowly in time, leading to a lower number of infected individuals at time $t = T$ with respect to **f** the same epidemics spreading on the aggregated network. The geographic data are provided by OpenStreetMap.

$M_{ij}^\alpha$ is observed}. Supposing that we have the aggregate topology $A^\mathcal{O}$ and the partial observation $\Gamma$ from a multiplex network $\mathbf{M}$ with $L$ layers, our goal is to predict whether there is a link between any two nodes $i, j$ in layer $\alpha$ when $(i, j, \alpha) \notin \Gamma$ (Fig. 2a). Once given an aggregate topology $A^\mathcal{O}$, there are in total $L \cdot |A^\mathcal{O}|$ links are expected to predict, where $|A^\mathcal{O}|$ indicates the number of edge in the aggregate topology $A^\mathcal{O}$. Notice that, among the $L \cdot |A^\mathcal{O}|$ potential links, we have already observed $card(\Gamma)$ links, where $card(\cdot)$ indicates the elements number of a set. Thus, we denote the fraction of partial observations by $c$ ($0 \le c < 1$), indicating the ratio between the number of observed edges in layers and all edges in the multiplex network, i.e., $c = \frac{card(\Gamma)}{L \cdot |A^\mathcal{O}|}$.

The first step of the reconstruction is to find the most probable value of $\theta$ by maximizing the posterior probability $P(\theta|A^\mathcal{O}, \Gamma)$, where $\theta$ is the network model parameter. Notice that the probability here provides a general description for any form of the network parameter $\theta$ (see Supplementary Note 2 for specific forms with a certain network model). Since there is no prior knowledge about the parameter $\theta$, we assume it to have a uniform distribution, i.e., $P(\theta) = const.$[33] In this case, based on the Bayesian rule, the maximum posterior estimate is equivalent to maximizing the log-likelihood function

$$l(\theta) = \ln P(A^\mathcal{O}, \Gamma|\theta), \qquad (2)$$

which performs the maximum likelihood estimation (MLE). The second step is to reconstruct the multiplex structure $\mathbf{M}$, inferring the probability for each possible structure specifically. Since many potential layer structures can produce the same multiplex aggregate topology, we denote the probability distribution for all multiplex structures by $Q(\mathbf{M})$, where $\sum_{\mathbf{M}} Q(\mathbf{M}) = 1$. Then, the estimated parameter $\theta$ can reconstruct the multiplex structure by calculating the posterior distribution

$$Q(\mathbf{M}) = P(\mathbf{M}|A^\mathcal{O}, \Gamma, \theta). \qquad (3)$$

However, there is a gap between the observations and network model parameters $\theta$, since the multiplex structure $\mathbf{M}$ is a hidden variable in $l(\theta)$, where $l(\theta) = \ln \sum_{\mathbf{M}} P(A^\mathcal{O}, \Gamma, \mathbf{M}|\theta)$. Notice that the sums over $\mathbf{M}$ here are expected to be over $\mathbf{M}$ that are consistent with $A^\mathcal{O}$ and $\Gamma$. For any distribution $Q(\mathbf{M})$, by employing the Jensen's inequality, we have

$$l(\theta) = \ln \sum_{\mathbf{M}} P(A^\mathcal{O}, \Gamma, \mathbf{M}|\theta) \ge \sum_{\mathbf{M}} Q(\mathbf{M}) \ln \frac{P(A^\mathcal{O}, \Gamma, \mathbf{M}|\theta)}{Q(\mathbf{M})}, \qquad (4)$$

where the equality holds if and only if the Eq. (3) is satisfied. Thus, $l(\theta)$ and $Q(\mathbf{M})$ are interdependent, and we perform an iterative process to obtain the MLE of the parameter $\theta$ and the posterior distribution $Q(\mathbf{M})$ as follows. Given an arbitrary initial value $\theta^{(0)}$, we find the optimized posterior distribution $Q^{(k)}(\mathbf{M})$ by Eq. (3). Then, we update the parameters $\theta^{(k)}$ that maximize the right-hand side of Eq. (4) by posterior distribution $Q^{(k-1)}(\mathbf{M})$, which performs a coordinate ascent to maximize the log-likelihood function (Fig. 2b). The iterations above are derived from the expectation-maximization (EM) algorithm[34] (details in the "Methods" section), and a toy example is shown (Fig. 2c). Note that if there is any prior on the parameters $\theta$, the proposed framework above can be improved by maximizing the product of the likelihood function $P(A^\mathcal{O}, \Gamma|\theta)$ and the prior $P(\theta)$, i.e., the so-called maximum a posterior estimation (MAP).

In estimation and statistics theory, an unbiased estimator is called efficient if the variance of the estimator reaches Cramer-Rao lower bound (CRLB)[35]. Fortunately, the proposed framework yields a maximum likelihood estimation, which is an unbiased estimator, and performs asymptotic normality indicating the estimator converges in distribution to a normal distribution[36]. With this, we prove that the variance of the estimator designed in our framework decreases as the fraction of partial observations $c$ increases, and further reaches the CRLB when the network size $N$ approaches infinity (see Supplementary Note 3 and Supplementary Fig. 4 for more details).

**Evaluations for the performance of the reconstruction.** We now analyze the performance of reconstruction on various real-world multiplex networks. Notice that the framework works for any given analytical generative model, such as Erdos-Rényi random network model and stochastic block model. For illustration, we

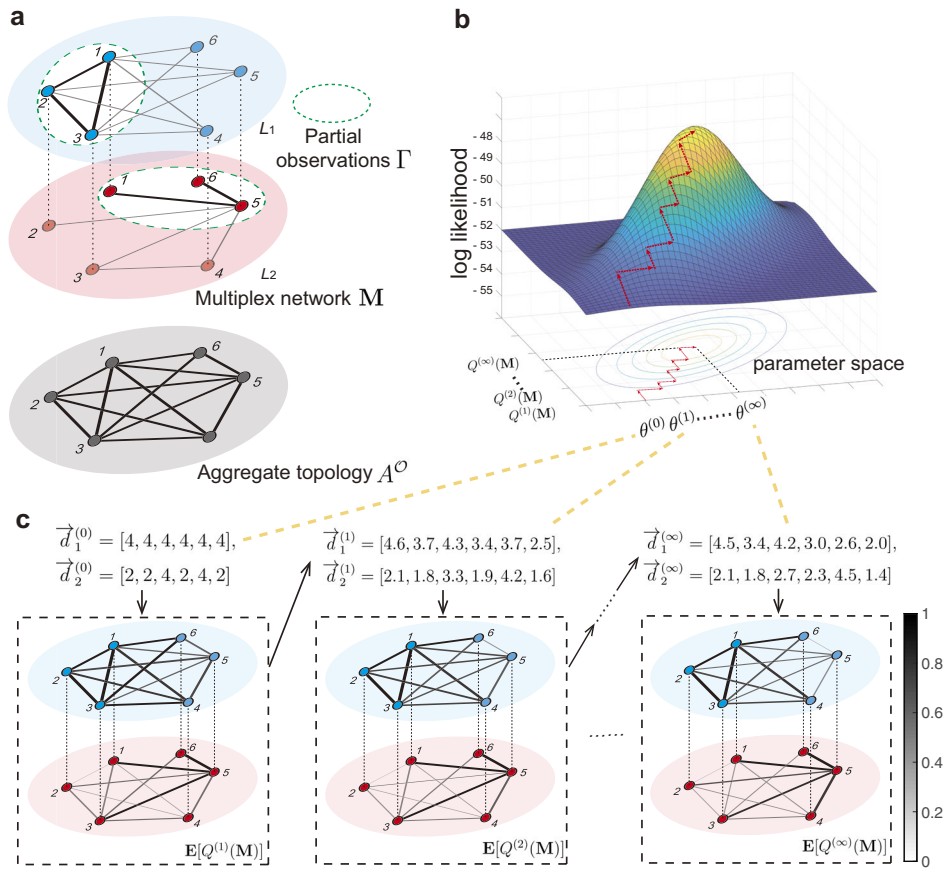

**Fig. 2 A schematic illustration of the reconstruction method for multiplex networks. a** A multiplex networks M is composed by $L_1$ and $L_2$ (the two layers of the multiplex network), and the bold edges in dashed white area indicate the partial observations $\Gamma$. $A^{\mathcal{O}}$ is the monoplex topology aggregated by the multiplex network M. The aggregate topology $A^{\mathcal{O}}$ and partial observations $\Gamma$ are leveraged to reconstruct the links in different layers that have not been observed directly. **b** The locus of the coordinate ascent method is shown in the probability space, where the red arrows show the repetitive process updating the parameter $\theta$ and structure distribution $Q(M)$ maximizes the likelihood. **c** A toy example is provided to demonstrate the specific iterative steps, where the parameter $\theta$ indicates the degree sequences $\vec{d}_1$ and $\vec{d}_2$ in two layers. Once given initial $\vec{d}_1^{(0)}$ and $\vec{d}_2^{(0)}$, we can obtain the expectation of the structure distribution $E[Q^{(1)}(M)]$, where the gray level and the thickness of each link indicates the existent probability estimated by the proposed method. When repeating this process, we can obtain the convergent degree sequences $\vec{d}_1^{(\infty)}$, $\vec{d}_2^{(\infty)}$, and the convergent expectation $E[Q^{(\infty)}(M)]$.

employ the configuration model as our network model, which exploits the parameter set $D$ to describe model parameters. For a multiplex network composed by $L$ layers, specifically, the parameter set $D$ contains $L$ vectors $\vec{d^\alpha} \in \mathbb{R}^N$ ($\alpha = 1, 2, \cdots, L$), encoding the degree sequences in each layer, where the component $d^\alpha(i)$ describes the degree of node $i$ in layer $\alpha$. The configuration model can significantly reduce the complexity from exponential to polynomial by exploiting the independence of each link and this model has been widely applied to analyze the relationship between structure and function of complex networks[37–39].

As we mentioned in the last section, once a certain network model is determined, we can conduct specific derivations to find the most probable values of all parameters in $D$ by maximizing the likelihood $P(A^{\mathcal{O}}, \Gamma|D)$ (see Supplementary Note 2 for detailed derivations). After estimating degree sequences $D$, the posterior probability $Q_{ij}^\alpha$ can be calculated by

$$Q_{ij}^\alpha = P(M_{ij}^\alpha = 1|A^{\mathcal{O}}, \Gamma, D), \quad (5)$$

which is called here link reliability, measuring the probability that a link exists between node $i$ and node $j$ in layer $\alpha$ (see Supplementary Note 2 for complete algorithm and complexity

analysis in this case). We examine the reliability of all links in the testing set $E^T$ consisting of potential links except those of the partial observations, i.e., $E^T = \{(i, j, \alpha) \notin \Gamma | A_{ij}^{\mathcal{O}} = 1\}$ (see Supplementary Fig. 5 for a schematic illustration). For this purpose, we calculate the TP (true positive rate) $P(M_{ij}^\alpha = 1|Q_{ij}^\alpha > q)$, FP (false positive rate) $P(M_{ij}^\alpha = 0|Q_{ij}^\alpha > q)$, TN (true negative rate) $P(M_{ij}^\alpha = 0|Q_{ij}^\alpha < q)$ and FN (false negative rate) $P(M_{ij}^\alpha = 1|Q_{ij}^\alpha < q)$ in $E^T$, where $q$ is the threshold that determines the classifier boundary for varying classes.

We first vary the threshold $q$ from 0 to 1, and calculate AUC (area under the receiver operating characteristic (ROC) curve) for two real-world datasets (C. elegans neural network and London transportation network) against different $c$. In the meantime, we compare our results with three-link prediction methods that performed well on inference tasks by partial observations so far. The first relevant work is referred to De Bacco et al., who have proposed a generative model, and an efficient expectation-maximization algorithm, which allows to perform inference tasks such as community detection and link prediction[40]. It works for multiplex networks with groups, but it may fail in networks without group-based structures. The second is referred to Tarres-Deulofeu et al., who introduced a stochastic block model, which

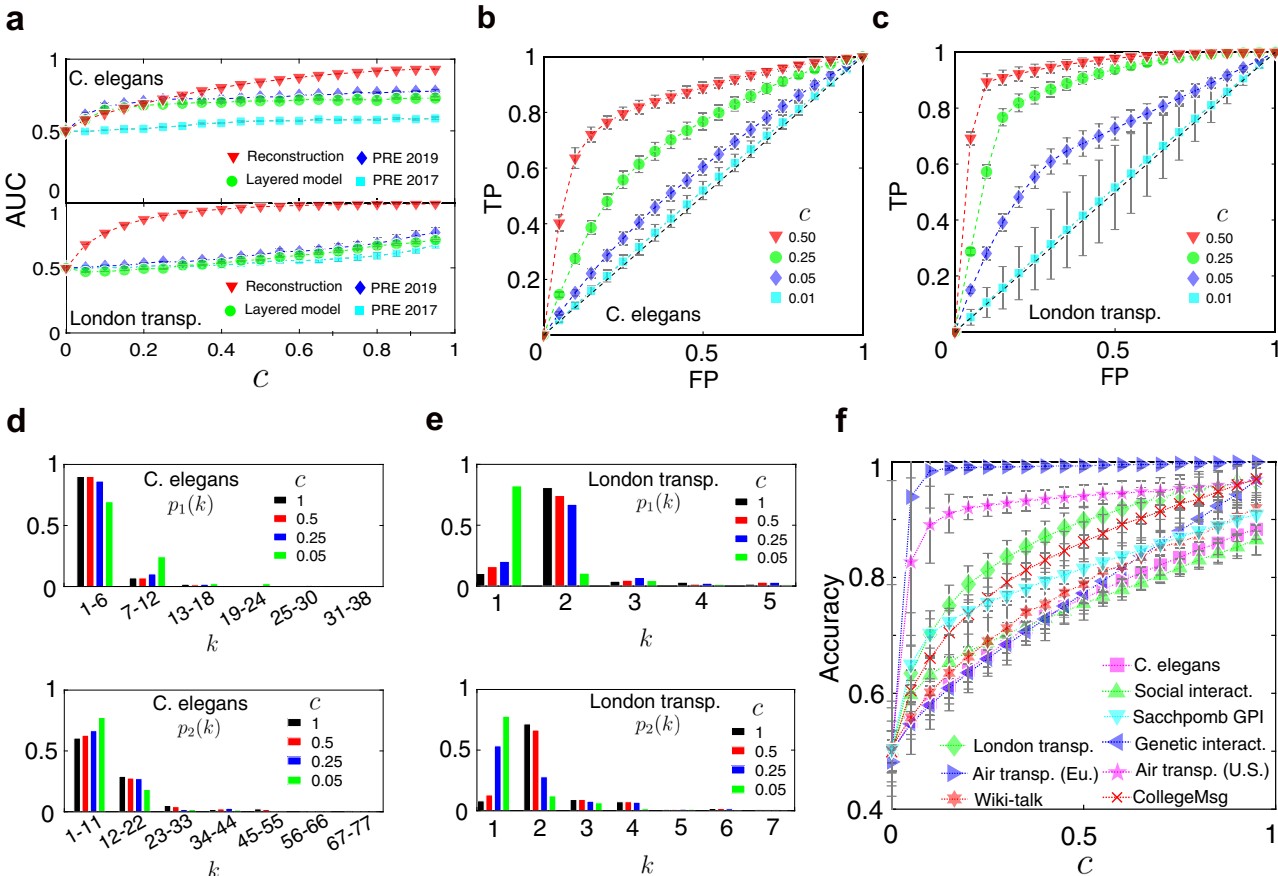

**Fig. 3 Performance of our reconstruction framework in various real multiplex networks.** We calculate the AUCs (area under the receiver operating characteristic curve) by four methods for C. elegans neural network and London transportation network in **a**, where the baselines refer to PRE 2019[41], PRE 2017[40], and Layered model (by a single-layer link prediction method, layer by layer, in the multiplex network). The ROC (receiver operating characteristic) curves for four specific values of the fraction of partial observations $c = 0.01, 0.05, 0.25, 0.5$ are shown in **b** and **c**, respectively. Here the x-axis denotes the false positive rate and the y-axis denotes the true positive rate. Increasing the threshold results in fewer false positives (and more false negatives), corresponding to moving left on the curve from the top right corners to the left bottom along the ROC curve, where a random guess gives a point along the dashed diagonal line. The inferred degree distributions for three values of $c$ are shown in **d** and **e** for the two real-world networks and compared to real ones ($c = 1$). The x-axis indicates the degree $k$ and the y-axis represents the probability that the degree of a randomly chosen node is equal to $k$. Further, for a specific value of $q = 0.5$, we compare the accuracy of the reconstructed networks using our proposed framework for nine real-world networks in **f** by increasing $c$ from 0 to 0.95. The error bar indicates the standard deviation in this figure.

can take full advantage of the information contained in the whole network[41]. The third baseline is calculated by a single-layer link prediction method, layer by layer, in the multiplex network (Layered model). Notice that all of them did not take the aggregate topology into consideration, while our method can provide valuable insights into how to aggregate topology information helps reconstruction. The AUCs by four methods for C. elegans neural network and London transportation are shown in Fig. 3a, displaying that the aggregate topology truly provides important information (see Supplementary Fig. 13 for more empirical results).

Further, we display the ROC curves in the ROC space for $c = 0.01, 0.05, 0.25, 0.5$, respectively (Fig. 3b, c). Here, the ROC space is defined by plotting the false positive rate on the x-axis and the true positive rate on the y-axis, displaying the relative trade-offs between false positive (costs) and true positive (benefits). Notice that the ROC curve describes the performance as a continuous function of the threshold $q$, and can well quantify the true positive rate and the false-positive rate for any given threshold $q$. Thus, AUC is a scalar that quantifies the performance and it does not depend on the choice of threshold. The results of the ROC analysis show that our proposed framework is very

effective for any threshold in the interval of $(0, 1)$, and thus it is stable for different thresholds in various real-world networks. Further, the true positive rate is positively correlated with the false-positive rate, and there exists a threshold, above which a false positive rate increases faster than the true positive rate. It is, thereby, not justified anymore to improve a true positive rate by increasing the false positive rate beyond such a threshold.

Next, we will set specific thresholds for further analysis. For example, since we do have any prior knowledge about the threshold $q$ in the task, we set $q = 0.5$ to avoid arbitrary choice, i.e., when the link reliability is larger than 0.5, an edge is considered to exist, and vice versa. Then, we calculate the four metrics to evaluate the performance of multiplex reconstruction for the nine real-world datasets. As a result, the fraction of partial observations $c$, indicating the portion of the observed edges, exhibits a positive correlation with the accuracy of the reconstruction, showing good performance even for a quite small $c$ (Fig. 3f) (see the "Methods" section and Supplementary Fig. 6 for more details for the evaluations). In practice, we sample the partial observations by vertex sampling, and repeat the reconstruction 100 times for each value of $c$ (see the "Methods" section for more details about the sampling of partial observations).

Besides link reliability, network characteristics also include average degree, degree distribution, length distribution of the shortest paths, which are significant to network reconstruction. One prominent advantage of the reconstruction framework is that we can simultaneously obtain other micro- or mesoscale network properties. For example, the expectation of the degree distribution in layer $\alpha$ can be obtained by $E(p_\alpha) = \sum_\mathbf{M} Q(\mathbf{M}) \cdot p_\alpha(\mathbf{M})$, where $p_\alpha(\mathbf{M})$ is the degree distribution in layer $\alpha$ for any specific multiplex structure $\mathbf{M}$. The reconstructed degree distributions in two layers with different values of $c$ are compared to the ground truth ($c = 1$), demonstrating that the degree distributions in all layers can also be well reconstructed as $c$ increases (Fig. 3d, e). Generally, the expectation of property $X$ is the first raw moment obtained by $E(X) = \sum_\mathbf{M} Q(\mathbf{M}) \cdot X(\mathbf{M})$, while the corresponding variance is the second central moment, $D(X) = \sum_\mathbf{M} Q(\mathbf{M}) \cdot [X(\mathbf{M}) - E(X)]^2$. Moreover, we can obtain skewness, kurtosis, and higher moments of a property $X$. We will discuss how different network characteristics (e.g., average degrees and overlap of edges in different layers) impact the performance of reconstruction in the next section.

**Reconstructability determined by the discrimination indicator**. It is of great significance to investigate how various characteristics of multiplex networks affect the reconstructability, i.e., the reconstruction accuracy we measured. Without loss of generality, we conduct an analysis of two-layer multiplex networks to illustrate the method in detail. Once a link is observed in the aggregate network, the probability space of the potential multiplex structure contains three events: the potential link exists (i) only in layer 1, (ii) only in layer 2, or (iii) in both layers. Then, the uncertainty of all links in the multiplex network $\mathbf{M}$ can be quantified by the information entropy

$$\mathcal{H}(\mathbf{M}|A^{\mathcal{O}}) = \mathcal{H}(M^1|A^{\mathcal{O}}) + \mathcal{H}(M^2|A^{\mathcal{O}}). \quad (6)$$

Generally speaking, the smaller the information entropy $\mathcal{H}$ is, the more certain the potential multiplex structure is, and vice versa.

To study how different characteristics of multiplex networks impact the information entropy, we first introduce the ratio of average degrees of two layers denoted by $r$, i.e., $r = \frac{\langle k_1 \rangle}{\langle k_2 \rangle}$, where $\langle k_1 \rangle$ and $\langle k_2 \rangle$ are the average degrees in layer 1 and layer 2, respectively. We assume, without loss of generality, that $\langle k_1 \rangle \leq \langle k_2 \rangle$, such that $0 < r \leq 1$. Then, we consider the overlap of edges denoted by $v$ between the two layers. A high overlap indicates that a link is more likely to exist in one layer if the corresponding link exists in the other layer, i.e., a low uncertainty. To measure the overlap $v$ of a multiplex network, we refer to the Jaccard index of $\mathcal{E}_1$ and $\mathcal{E}_2$ that indicate the two edge sets in the two layers, i.e., $v = |\mathcal{E}_1 \cap \mathcal{E}_2| / |\mathcal{E}_1 \cup \mathcal{E}_2|$ (see Supplementary Fig. 9 for more details about multiplex network characteristics).

We next explore how these factors impact the information entropy of a multiplex network and further determine the performance of reconstruction. By Eq. (6), we can calculate the information entropy $\mathcal{H}$ with a mean-field approximation, and obtain

$$\mathcal{H}(\mathbf{M}|A^{\mathcal{O}}) = -|A^{\mathcal{O}}| \cdot \sum_{\alpha=1}^{2} [\bar{p}_\alpha \cdot \ln \bar{p}_\alpha + (1 - \bar{p}_\alpha) \cdot \ln(1 - \bar{p}_\alpha)], \quad (7)$$

where

$$\bar{p}_1 = E[P(M_{ij}^1 = 1|A_{ij}^{\mathcal{O}} = 1)] = \frac{\hat{v} + \hat{r}}{1 + \hat{r}}, \quad (8)$$

and

$$\bar{p}_2 = E[P(M_{ij}^2 = 1|A_{ij}^{\mathcal{O}} = 1)] = \frac{1 + \hat{v} \cdot \hat{r}}{1 + \hat{r}}, \quad (9)$$

indicating the average probability for the existence of any link in layer 1 and layer 2, respectively. Notice that in Eqs. (8) and (9), $\hat{v}$ and $\hat{r}$ are the estimations of $v$ and $r$ when we only have partial observations $\Gamma$, and we approximate them by $c \cdot v$ and $r^c$ empirically (see Methods section for more details). Thus, we find that the information entropy of a given multiplex network is highly related to the fraction of partial observations $c$, the ratio of average degrees $r$ and overlap $v$. It is clear that the uncertainty of the probability space decreases with the increasing of $c$ and $v$. Hence, the information entropy $\mathcal{H}$ is a monotonously decreasing function of $c$ and $v$ over the domain. For $r$, however, the information entropy is a monotonously increasing function when $r$ increases from 0 to 1 (Fig. 4a). Clearly, $\mathcal{H}$ describes the microscale discrimination between layers of a multiplex network, since a high discrimination (e.g., $r$ tends to 0) leads to a low information entropy.

Generally, the reconstruction accuracy is expected to be determined by the information entropy $\mathcal{H}$, since $\mathcal{H}$ is the primary variable that affects the uncertainty of the distribution of potential multiplex structures. Empirically, we find that the accuracy is linearly determined by the indicator $\rho \cdot \mathcal{H}$ (Fig. 4b), i.e.,

$$\text{Accuracy} \approx 1 - \rho \cdot \mathcal{H}, \quad (10)$$

where $\rho$ is a scaling factor satisfying

$$\rho = \frac{1}{2 \ln 2 \cdot |A^{\mathcal{O}}|} \cdot \left(1 - \frac{1 - v}{1 + v} \cdot c^s\right). \quad (11)$$

In Eq. (11) above, $s = s_{(\mathbf{M})}$ is a constant related to the topology of the multiplex network $\mathbf{M}$ (see Supplementary Table 2 for approximate values of $s_{(\mathbf{M})}$. The term $(1 - v)/(1 + v)$ in Eq. (11) indicates the uncertainty of links in the testing set can be reduced by partial observations. The parameter $s$ describes the scale how partial observations can reduce the uncertainty of links in the testing set (details in Methods section). We further find that $s_{(\mathbf{M})}$ is closely proportional to the cosine similarity of the two degree sequences in each layer, i.e., $s \propto \cos\langle \overrightarrow{d^1}, \overrightarrow{d^2} \rangle$ (Fig. 4c). Clearly, $\cos\langle \overrightarrow{d^1}, \overrightarrow{d^2} \rangle$ describes the similarity between degree sequences of the two layers in a multiplex network, indicating the mesoscale discrimination, which is not relevant to microscale discrimination including $r$, $v$, and $\mathcal{H}$ generally (Fig. 4d).

Thus, the reconstructability is determined by the discrimination indicator $(1 - \rho \cdot \mathcal{H})$ from both microscale and mesoscale views. This discovery indicates that the reconstruction can be predicted accurately by the discrimination indicator, obtaining a high accuracy of reconstruction where either $\rho$ or $\mathcal{H}$ is small. For example, the reconstructability can be enhanced when the difference in average degrees between layers is vast ($r$ tends to 0). Notice that we can approximate $s$ by the cosine similarity if we do not meet the exact value of $s$ empirically, since $s$ is highly related to the cosine similarity. We will next discuss how to allocate the partial observations in different layers when a specific budget $\bar{c}$ is given.

**Allocating limited budget for partial observations**. Usually, we have a limited budget for conducting observations in practice. It is, thereby, necessary to investigate budget allocation (partial observations $\Gamma$) in different layers to optimize the performance of reconstruction (e.g., the accuracy) as far as possible. We denote the average fraction of partial observations in each layer by $\bar{c}$, i.e., $\bar{c} = \sum_\alpha c_\alpha / L$, where $c_\alpha$ indicates the fraction of partial observation

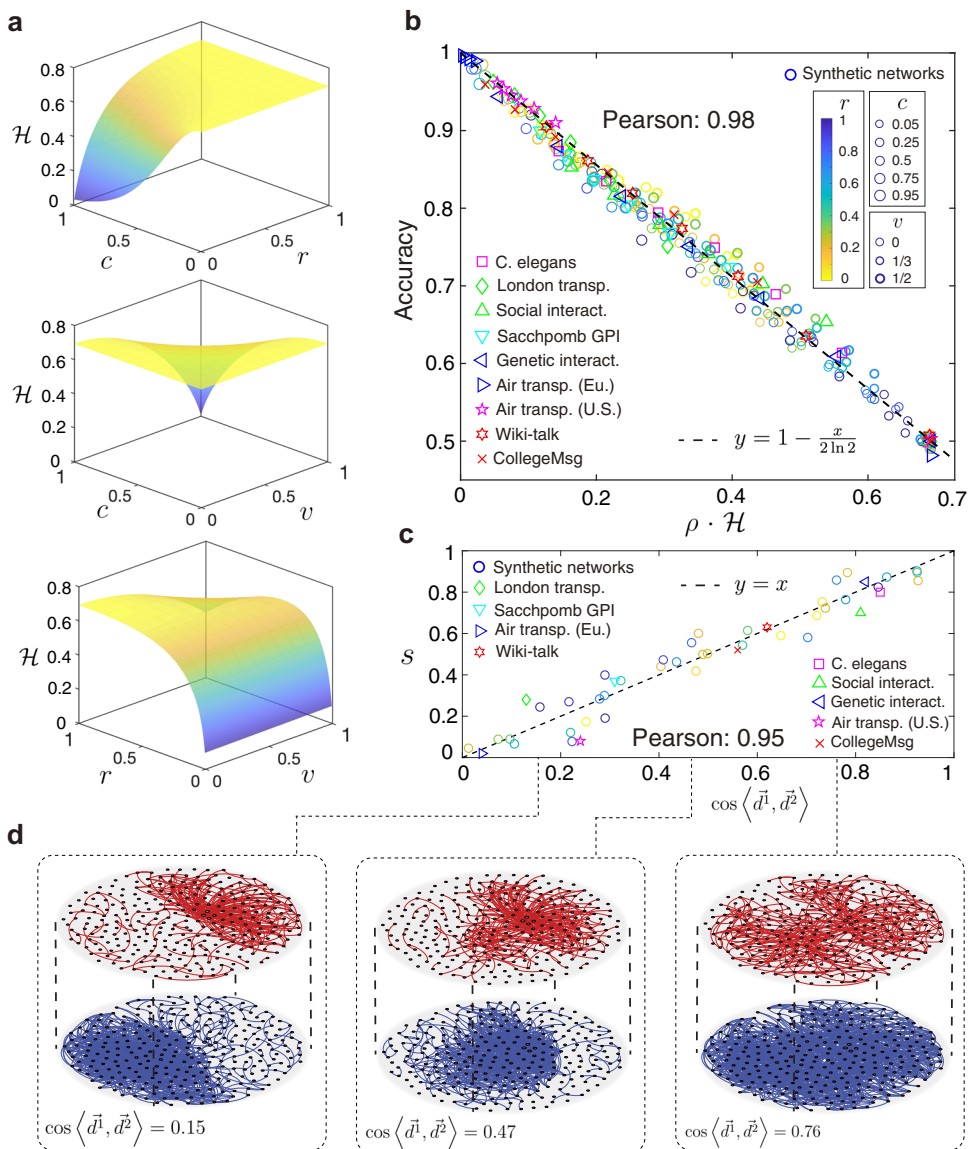

**Fig. 4 The impact of multiplex network characteristics on reconstruction. a** The discrimination indicator for reconstruction is influenced by $c$ (fraction of partial observations), $r$ (ratio of average degrees), and $v$ (overlap of edges). **b** The relationship between accuracy of reconstruction and the discrimination indicator ($\rho \cdot \mathcal{H}$) for nine real-world networks and several synthetic networks in two dimension, showing the discrimination indicator is a good predictor (Pearson correlation is 0.98) for accuracy of reconstruction. **c** The correlation (Pearson correlation is 0.95) between the parameter $s_{(M)}$ and the cosine similarity of two degree sequences, $\cos\langle \vec{d^1}, \vec{d^2} \rangle$, for nine real-world networks and several synthetic networks in two dimension, where the parameter $s$ for different multiplex networks are shown in Supplementary Table 2. **d** Three synthetic networks corresponding to the degree sequence $\cos\langle \vec{d^1}, \vec{d^2} \rangle = 0.15, 0.47, 0.76$, respectively.

in layer $\alpha$, and denote $\mathcal{H}(M^\alpha|A^{\mathcal{O}})$ by $\mathcal{H}_\alpha$ for simplicity. Similarly, employing the mean-field approximation (details in the "Methods" section), we can predict the accuracy by the function $F$ defined as

$$F(c_1, c_2) = 1 - \frac{1-c_1}{1-c_1+(1-c_2)/\hat{r}} \cdot \rho_1 \cdot \mathcal{H}_1 - \frac{(1-c_2)/\hat{r}}{1-c_1+(1-c_2)/\hat{r}} \cdot \rho_2 \cdot \mathcal{H}_2, \quad (12)$$

where

$$\rho_\alpha = \frac{1}{2\ln 2 \cdot |A^{\mathcal{O}}|} \cdot \left(1 - \frac{1-v}{1+v} \cdot c_{3-\alpha}^s\right), \quad \alpha = 1, 2. \quad (13)$$

We next explore how the performance of reconstruction is impacted by the ratio $c_1/c_2$ when a certain budget is given for partial observations. Once $\bar{c}$ is given, we regard the function $F$ as a unary function of $c_1$, i.e., $F = F(c_1)$, since $c_2 = 2\bar{c} - c_1$. Then, the domain of $F(c_1)$ is $[0, 2\bar{c}]$ if $\bar{c} \leq 0.5$, and is $[2\bar{c} - 1, 1]$ if $\bar{c} > 0.5$.

We notice that the function $F(c_1)$ is monotonically increasing over the domain if $\bar{c}$ is small, but decreases at first and increases later if $\bar{c}$ is large. Theoretical analysis shows that $F(0) \geq F(2\bar{c})$ if $\bar{c} \leq 0.5$, and $F(2\bar{c} - 1) \geq F(1)$ if $\bar{c} > 0.5$ (details in the "Methods" section). The result indicates that it is always better to allocate the budget as much as possible to the layer whose average degree is lower, and we can reach the optimal strategy to obtain the highest accuracy for the given network model then. Moreover, there exists a threshold $0 < \bar{c}_0(M) < 1$ for each multiplex network $M$,

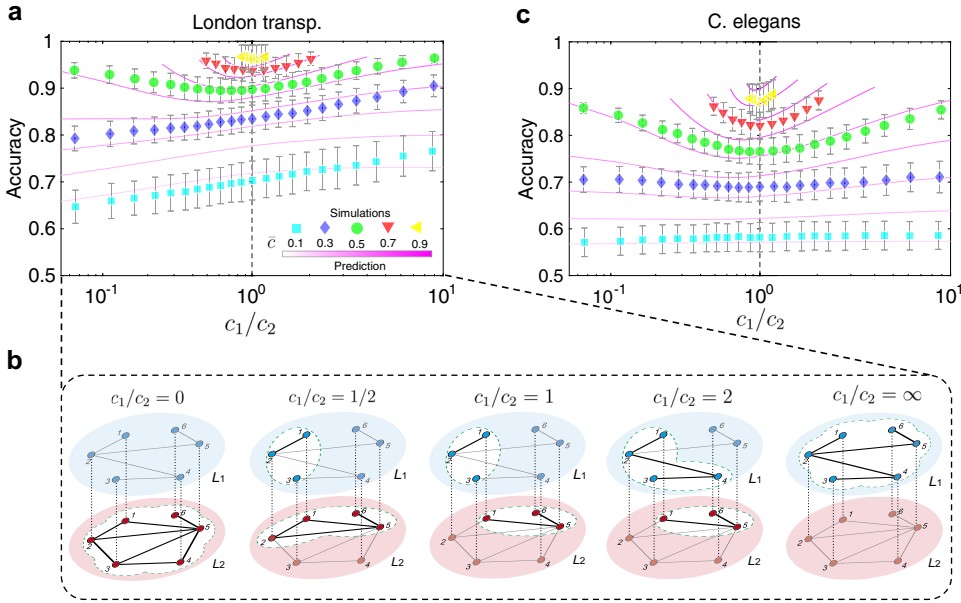

**Fig. 5 Allocating budget for reconstruction. a** The accuracy of reconstruction as a function of $c_1/c_2$ from 0 to ∞ for the London transportation multiplex network, where $c_1$ and $c_2$ indicate the fractions of partial observations in layer 1 and layer 2, respectively. The colored symbols indicates the simulation results corresponding to different budgets $\bar{c}$, while the magenta lines are the predictive results based on the discrimination indicator. **b** Different strategies to allocate limited budget are illustrated in five toy multiplex networks, where the dashed white area is the partial observations of multiplex networks. **c** The accuracy of reconstruction as a function of $c_1/c_2$ from 0 to ∞, when given total budget $\bar{c}$ for the C. elegans multiplex connectome. The error bar indicates standard deviation in this figure.

where $\bar{c}_0$ is the solution to the equation

$$F(0) = F(2\bar{c}_0). \tag{14}$$

If the budget $\bar{c}$ is less than $\bar{c}_0$, the accuracy increases when $c_1/c_2$ increases, and reaches the maximum as $c_1/c_2$ tends to ∞. If the budget $\bar{c}$ is large ($\bar{c} > \bar{c}_0$), however, the accuracy increases when $c_1/c_2$ tends to 0 or ∞, and reaches the maximum as $c_1/c_2$ tends to ∞ (Fig. 5a), indicating that the multiplex network can be reconstructed when the aggregate topology and either of the two layers is observed (Fig. 5b). The reason is as follows. The partial observations in different layers can capture the maximal characteristics of each layer when $c_1/c_2 = 1$. However, it will lead to more redundancy and lower accuracy if the partial observations in different layers have a high overlap of observations, making the performance even worse when $c_1/c_2 = 1$ and $\bar{c}$ is large. The theoretical results enable us to make the best strategy to allocate budget and thus obtain the optimal reconstruction of multiplex networks in this case. Furthermore, results from real-world data sets verified our theoretical analysis (Fig. 5a, c). We will discuss how different multiplex network characteristics impact the performance of reconstruction from a dynamical behavior point of view in the next section.

**Predicting dynamic processes in multiplex networks**. We proceed to investigate the performance of the reconstructed multiplex networks on the prediction of dynamic processes, which is critical to the network functionality. First, we study a percolation process occurring on a two-layer interdependent multiplex network. In such a multiplex network, once a set of nodes is removed (e.g., being attacked or random failure) in one layer, nodes disconnected to the GCC (giant connected component) in the same layer and the counterparts of the removed nodes will also fail and thus be removed. The new removed nodes result in more node removal, and the repetitive processes lead to the catastrophic cascade of failures. For the reconstructed multiplex network encoded by the expectation $E[Q(M)]$, we binarize the matrix

$E[Q(M)]$ and randomly remove nodes in one layer with probability $1 - p$ (see Supplementary Note 4 for more details of the process). We calculate the size of GMCC (giant mutually connected component) as a function of $p$ and the critical threshold $p_c$, above which the GMCC exists. We compare the average size of GMCC in the reconstructed network (repeated 100 times) to the real one with the C. elegans neural network against ranging $c$ (Fig. 6a). The performance of reconstruction is well as seen from the size of GMCC, even if $c$ is small. The estimates of the size of GMCC and the critical probability $p_c$ approach those of the real networks ($c = 1$) as $c$ increases 1. However, the proposed method slightly underestimates both the size of GMCC and the critical threshold $p_c$ for the C. elegans neural network. Further, simulations on synthetic networks reveal that the method underestimates much more the robustness and $p_c$ of the interdependent networks when $r$ is small (Fig. 6b). When $r$ is small and close to 0, the edges in the multiplex network are concentrated on one layer, leading to the extreme vulnerability. However, the reconstructed method could not capture the unbalance, especially for a small fraction of partial observation.

Second, we consider a random walk process taking place on interconnected multiplex networks, where interlayer links only exist between counterparts. We suppose that a number of walkers start from randomly chosen nodes and walk along with intralayer links with a probability $p_{intra}$, and along with interlayer links with probability $p_{inter}$ (see Supplementary Note 5 for more details). We employ the coverage $\phi(t)$ as the performance metric, indicating the fraction of nodes that have been visited by the walkers before time $t$. The coverage at each time on reconstructed multiplex networks are compared to the real one (London multiplex transportation network) against different $c$, showing an outstanding good prediction as $c$ increases (Fig. 6c). Simulations on synthetic networks show that the coverage will be overestimated no matter $r$ is small or large (Fig. 6d). Moreover, we conduct more simulations with real-world multiplex networks with more than two layers. For example, we simulated the random walk

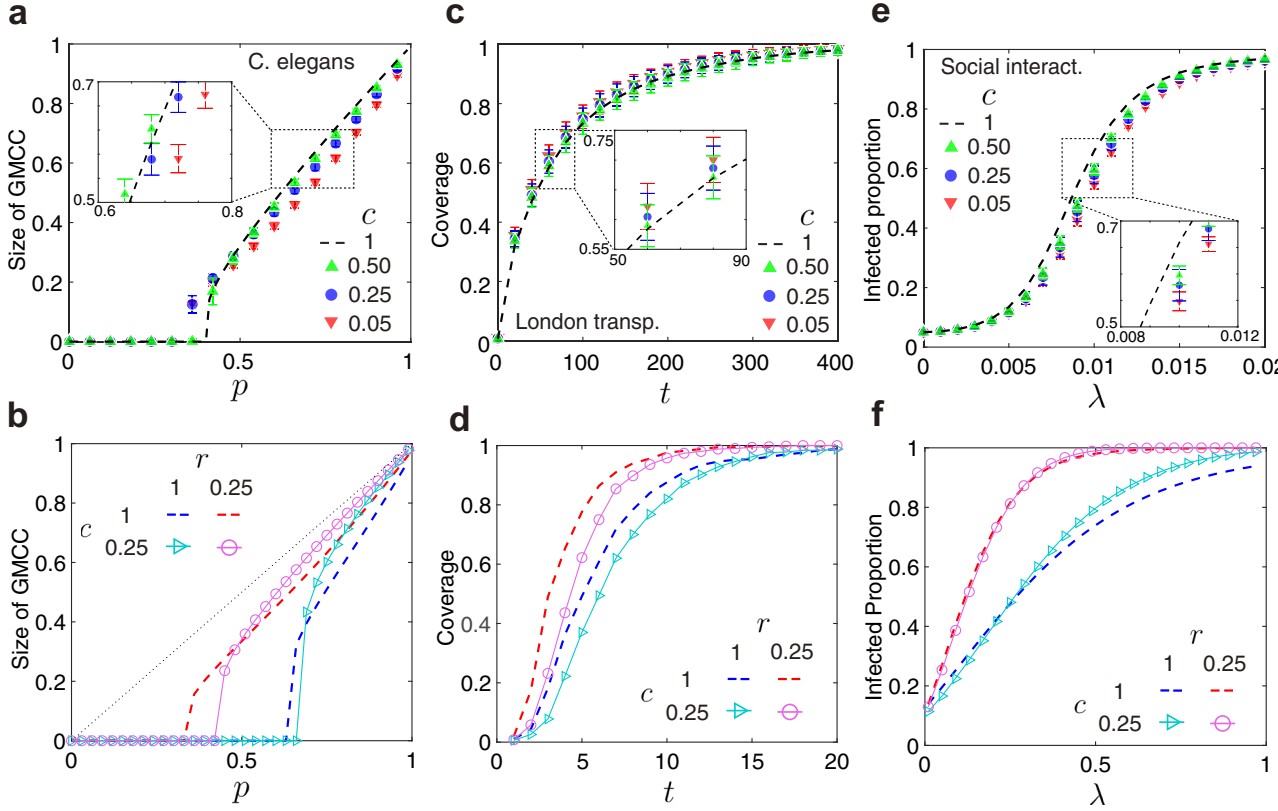

**Fig. 6 The performance of dynamic prediction. a** The percolation processes of the reconstructed multiplex network corresponding to different fraction of partial observation $c = 0.05, 0.25, 0.5$, and the real multiplex network ($c = 1$) for C. elegans multiplex connectome. The x-axis denotes the occupied probability $p$ and the y-axis denotes the size of GMCC (giant mutually connected component) when nodes are randomly removed with probability $1 - p$ in one layer. **b** The impact of ratio of average degrees $r$ on the percolation process for $r = 0.25$ and $r = 1$. **c** A random walk process taking place on the reconstructed multiplex network and real multiplex network for London transportation network. The x-axis denotes time $t$ and the y-axis denotes coverage (the fraction of nodes that have been visited before a certain time) of $n$ walkers starting from a set of random chosen nodes. **d** The impact of $r$ on random walk process for $r = 0.25$ and $r = 1$. **e** The spreading process on the reconstructed temporal network and real temporal network for the social interactions at the SFHH conference. The x-axis denotes the infection rate $\lambda$ and the y-axis denotes the infected fraction. **f** The impact of $r$ on spreading process for $r = 0.25$ and $r = 1$. The error bar indicates standard deviation in this figure.

processes on the European air transportation network which has three layers (Ryanair, Lufthansa, EasyJet), and the air transportation network in the U.S. which has four layers (SkyWest, Southwest, American Eagle, American Airlines) (see Supplementary Fig. 10c, d for more results on real-world networks).

Last, we investigate a spreading process based on the SI (susceptible-infected) model on temporal networks, where interlayer links only exist between two counterparts at two adjacent time slots[23]. In the SI model, each node has only two states: "susceptible" (S) or "infected" (I), and 5% nodes are randomly chosen to be sources (infected) at initial time $t = 0$ in our simulations. At each time (which corresponds to one layer in the temporal network), the infected nodes will infect the susceptible neighboring nodes with a specific infection rate $\lambda$ (see Supplementary Note 6 for more details). In the spreading process, the fraction of infected nodes $I(T)$ at time $t = T$ on the reconstructed networks is calculated and compared to the real one with the network of social interactions at SFHH (La Société française d'Hygiène Hospitalière) conference (Fig. 6e). Simulations on synthetic networks reveal that the method overestimates $I(T)$ when $r$ is close to 1 (Fig. 6f). We also compare these results with the existing methods (see Supplementary Fig. 11 for more details). Further, we simulated the spreading processes on the Wiki-talk network which has 14 layers (14 weeks), and the CollegeMsg network that has 22 layers (22 days) (see Supplementary Fig. 10e, f for more results on real-world networks).

Moreover, we have studied how the performance of reconstruction for dynamics is influenced by more network characteristics, including the overlap of edges and ratio of heterogeneity (see Supplementary Fig. 12 for more results on synthetic network with different characteristics).

## Discussion

Network reconstruction has attracted much research attention recently[24,31,42], since it has wide applications such as link prediction, community detection and systems' vulnerability analysis. Most previous studies focused on monoplex networks, and therefore there is a pressing need to develop a reconstruction framework for multiplex networks. Existing work have met success to determine if an observed monoplex network is the outcome of a hidden multilayer process by assuming generative models for each layer of the multiplex network. However, it is necessary to further explore the multiplex layers structure and predict the dynamics once it is verified that there is a hidden multiplex structure. Our primary goal is to reconstruct the multiplex structure from the knowledge of both an observed aggregate monoplex network and partial observations.

However, there are many challenges preventing us to build a framework for the reconstruction of multiplex networks. Given the aggregation mechanism (e.g., the OR mechanism), apparently, there are a large number of potential structures given the same

aggregate network. To avoid the ergodic methods, we propose a framework by building a probability space $Q(\mathbf{M})$, and reduce the complexity from exponential to polynomial by employing the configuration model that allows an arbitrary degree sequence. Generally, priors on generating models or other dynamic information are almost unavailable, while the local information (referred to as partial observations in this article) in some specific layers is more accessible. For this purpose, we need to estimate the network model parameter $\theta$ based on very limited partial observations and, unfortunately, it is interdependent on the posterior probability distribution $Q(\mathbf{M})$. We design here an efficient mathematical framework based on the Expectation-Maximization method performing a maximum likelihood estimation, and prove that the variance of the estimation reaches the CRLB when network size $N$ approaches infinity. We evaluate the performance of the reconstruction using various empirical multiplex networks, ranging from microscale (e.g., the accuracy of link reliability) to mesoscale (e.g., degree distributions). Empirical results demonstrate that the performance of reconstruction mounts quickly initially with a small fraction of partial observations, exhibiting the power of the proposed reconstruction framework. Applying mean-field approximation, we find that a discrimination indicator that integrates all considerable factors determines the accuracy of the reconstruction, which theoretically aids us to have a deep understanding the relation between the reconstructability and the network characteristics. Thus, the indicator enables us to make the best strategy to allocate limited budget and obtain the highest accuracy for the given network model, i.e., the optimal reconstruction. Notice that though our analysis was illustrated by the configuration model that was based on the degree sequence, we found that the empirical results did not rely on the degree sequence. Finally, we investigate the performance from a dynamical view, finding that our proposed framework can well predict dynamic processes taking place on multiplex networks, and the impact of network characteristics (e.g., average degree, heterogeneity and overlap of edges in different layers) on performance is analyzed.

To the best of our knowledge, we provide here a comprehensive mathematical framework for reconstructing a multiplex network layer structures although there exist several good related works. For example, Bianconi et al. studied an ensemble consisting of all multiplex networks submitted to a given degree sequence, which is a constraint in their case[43]. In our work, however, the degree sequence is the parameter (of the configuration model) to be estimated, which is a fundamental distinction between our work and Bianconi's case. Thus, our framework paves insight on the relation between the structure and the reconstructability of multiplex networks, and our finding reveals the essential features of multiplex network reconstruction. In future work we will focus on improving the reconstruction performance by data other than node-pair interactions that can be helpful in terms of reconstructing multiplex networks. We took the configuration model as an illustration for our framework, since it is a relative general model that allows an arbitrary degree sequence. However, it remains necessary to study how well does the framework perform with group-based models or centrality-based models. Note that it is also interesting to reconstruct multiplex networks through the dynamics and steady state of the system[44,45]. We believe that our proposed framework will have an impact in many different applications including link prediction, missing links recovery, spurious links location drawn from biological, social, transportation domains.

## Methods

**Expectation maximization framework**. In this section, we will present details on how to obtain the maximum likelihood estimation of network model parameter $\theta$.

The primary objective is to find $\theta$ that maximizes the likelihood function $P(A^{\mathcal{O}}, \Gamma|\theta)$. In practice, we will maximize its logarithm $l(\theta) = \ln P(A^{\mathcal{O}}, \Gamma|\theta)$ rather than $P(A^{\mathcal{O}}, \Gamma|\theta)$ for computational convenience. Clearly, by employing the law of total probability, we have

$$l(\theta) = \ln P(A^{\mathcal{O}}, \Gamma|\theta) = \ln \sum_{\mathbf{M}} P(A^{\mathcal{O}}, \Gamma, \mathbf{M}|\theta), \qquad (15)$$

which is a summation over all possible potential multiplex structure $\mathbf{M}$. Employing the Jensen's inequality then, we have

$$l(\theta) = \ln \sum_{\mathbf{M}} P(A^{\mathcal{O}}, \Gamma, \mathbf{M}|\theta) \geq \sum_{\mathbf{M}} Q(\mathbf{M}) \ln \frac{P(A^{\mathcal{O}}, \Gamma, \mathbf{M}|\theta)}{Q(\mathbf{M})}, \qquad (16)$$

where the inequality holds as long as $Q(\mathbf{M})$ is a distribution of the multiplex structure $\mathbf{M}$ satisfying $\sum_{\mathbf{M}} Q(\mathbf{M}) = 1$. For simplicity, we denote the right-hand side of equation (16) by $J(Q, \theta)$, i.e.,

$$J(Q, \theta) = \sum_{\mathbf{M}} Q(\mathbf{M}) \ln \frac{P(A^{\mathcal{O}}, \Gamma, \mathbf{M}|\theta)}{Q(\mathbf{M})}. \qquad (17)$$

Notice that $J$ is a function of the distribution $Q(\mathbf{M})$ and the parameter $\theta$ simultaneously, and it is obviously a lower-bound function of the log-likelihood function $l(\theta)$.

In the expectation-maximization (EM) algorithm[34], we will maximize the lower-bound function $J$ by recursively executing two steps: E-step and M-step. In the E-step, we maximize $J(Q, \theta)$ while keeping $\theta$ constant. It is easy to see that the equality in Eq. (16) holds if and only if $Q(\mathbf{M})$ is the posterior distribution of the multiplex structure $\mathbf{M}$, i.e.,

$$Q(\mathbf{M}) = \frac{P(A^{\mathcal{O}}, \Gamma, \mathbf{M}|\theta)}{\sum_{\mathbf{M}} P(A^{\mathcal{O}}, \Gamma, \mathbf{M}|\theta)} = \frac{P(A^{\mathcal{O}}, \Gamma, \mathbf{M}|\theta)}{P(A^{\mathcal{O}}, \Gamma|\theta)} = P(\mathbf{M}|A^{\mathcal{O}}, \Gamma, \theta). \qquad (18)$$

In the M-step, we differentiate Eq. (17) with respect to $\theta$ while fixing $Q(\mathbf{M})$, and find the solution to the following equation

$$\frac{\partial}{\partial \theta} \sum_{\mathbf{M}} Q(\mathbf{M}) \ln P(A^{\mathcal{O}}, \Gamma, \mathbf{M}|\theta) = 0. \qquad (19)$$

Notice that the item $\sum_{\mathbf{M}} Q(\mathbf{M}) \ln P(A^{\mathcal{O}}, \Gamma, \mathbf{M}|\theta)$ is the posterior expectation of the log-likelihood function $\ln P(A^{\mathcal{O}}, \Gamma, \mathbf{M}|\theta)$ with respect to the distribution $Q(\mathbf{M})$. Thus, given an arbitrary initial value denoted by $\theta^{(0)}$, we can iteratively update the distribution $Q(\mathbf{M})$ and the parameter $\theta$ until convergence. The two steps can be written as the iteration scheme

$$\begin{cases} Q^{(k)}(\mathbf{M}) &= P(\mathbf{M}|A^{\mathcal{O}}, \Gamma, \theta^{(k)}). \\ \theta^{(k+1)} &= \arg\max_{\theta} E_{M \sim Q^{(k)}} \left[\ln P(A^{\mathcal{O}}, \Gamma, \mathbf{M}|\theta)\right]. \end{cases} \qquad (20)$$

Next we will briefly prove that the seris $\{\theta^{(k)}\}$ converges to the value that maximizes the log-likelihood. On one hand,

$$l(\theta^{(k+1)}) = \ln P(A^{\mathcal{O}}, \Gamma|\theta^{(k+1)}) = \ln \sum_{\mathbf{M}} P(A^{\mathcal{O}}, \Gamma, \mathbf{M}|\theta^{(k+1)}) \qquad (21)$$

$$\geq \sum_{\mathbf{M}} P(\mathbf{M}|A^{\mathcal{O}}, \Gamma, \theta^{(k)}) \ln \frac{P(A^{\mathcal{O}}, \Gamma, \mathbf{M}|\theta^{(k+1)})}{P(\mathbf{M}|A^{\mathcal{O}}, \Gamma, \theta^{(k)})} \qquad (22)$$

$$\geq \ln P(A^{\mathcal{O}}, \Gamma|\theta^{(k)}) \qquad (23)$$

$$= l(\theta^{(k)}) \qquad (24)$$

We can see that the sequence $\{l(\theta^{(k)})\}$ monotonously increases as $k$ grows. On the other hand, the log-likelihood sequence $\{l(\theta^{(k)})\}$ obviously has a upper bound, i.e., $\max \ln P(A^{\mathcal{O}}, \Gamma|\theta)$. Then $\{\theta^{(k)}\}$ converges to the value that loccally maximizes the log-likelihood function $l(\theta)$[46]. In practice, however, the log-likelihood function may have more than one local maximum values in more complex situations, while the EM algorithm is not guaranteed to converge to the global one. To overcome the problem, we try different random initial values for the parameter repeatedly, and find the global maximum of the likelihood value when they converge[47].

**Evaluation indices for reconstruction**. Accuracy, precision, recall, and AUC (area under the receiver operating characteristic curve) have been widely adopted to evaluate classification methods[48]. Accuracy is defined by the fraction between true results (both true positives and true negatives) and the total number of tests, i.e., accuracy $= (\text{TP} + \text{TN})/(\text{TP} + \text{TN} + \text{FP} + \text{FN})$; Precision gives the probability that a link exists in real network when reliability $Q^{\alpha}_{ij} > q$, i.e., precision $= \text{TP}/(\text{TP} + \text{FP})$; Recall equals to the proportion of true positive rate over true positive rate and false-negative rate, i.e., recall $= \text{TP}/(\text{TP}+\text{FN})$. In addition, for those links whose reliability $Q^{\alpha}_{ij} = q$, they have half contribution to the proportion. The AUC quantifies the expectation that the proposed method ranks a positive one higher than a negative one. Thus, all the tested links are ranked decreasingly according to their values of reliability, and the probability that a real link has a higher reliability than a nonexistent link is calculated. For imbalanced datasets (networks always tend to be sparse), the area under the precision-recall curve (AUPRC) is also a meaningful metric. Moreover, the Matthews correlation coefficient (MCC) is another popular choice, which produces a more informative and truthful score in evaluating binary

classifications[49]. These six metrics are used to evaluate the proposed framework for the nine real-world data sets (Supplementary Figs. 6 and 7).

**Partial observation sampling.** In this section, we would clarify the sampling of partial observations in our simulations. We sampled them by vertex sampling for a given fraction $c$. Specifically, for a large undirected network, there are $\frac{N^2}{2}$ pairs of nodes, where $N$ is the number of nodes. Note that, once we have observed these $\frac{N^2}{2}$ pairs, the network is certain and we do not need to infer the edge probability any more. Then, we randomly choose $n = \sqrt{c} \cdot N$ nodes, and the subgraph consisting of these nodes includes $\frac{(\sqrt{c} \cdot N)^2}{2} = c \cdot \frac{N^2}{2}$ edges in the original network. For each value of $c$, we repeated the reconstruction 100 times to avoid randomness, and drawn error bars in relevant figures of the manuscript.

**Mean-field approximation.** Our goal in this section is to clarify the discrimination indicator introduced in Results section by mean-field approximation. First we will calculate the information entropy $\mathcal{H}(\mathbf{M}|A^{\mathcal{O}})$ determined by the ratio of the average degrees $r$, overlap of edges $v$, and fraction of partial observations $c$.

Notice that $r$ ratio of average degrees of two layers and $v$ is the overlap of the edge sets in two layers, and they are not available when we only have partial observations, and thus they are estimated by $\hat{r}(r, c)$ and $\hat{v}(v, c)$. Since the average degree $\langle k_\alpha \rangle$ is a mesoscale property, we estimate $\langle \hat{k}_\alpha \rangle$ by $E[\langle k_\alpha \rangle] = \sum_{\mathbf{M}}[Q(\mathbf{M}) \cdot \langle k_\alpha \rangle(\mathbf{M})]$, indicating the expectation of $\langle k_\alpha \rangle$ for all potential multiplex structure. The results shown in Supplementary Fig. 8 allow us to estimate $\langle \hat{k}_\alpha \rangle$ approximately by

$$\left\langle \hat{k}_1 \right\rangle = \frac{\langle k_1 \rangle + \langle k_2 \rangle}{2} + \left[1 - (1 - \sqrt{c})^{2/r}\right] \cdot \frac{\langle k_1 \rangle - \langle k_2 \rangle}{2},$$
$$\left\langle \hat{k}_2 \right\rangle = \frac{\langle k_1 \rangle + \langle k_2 \rangle}{2} - \left[1 - (1 - \sqrt{c})^{2/r}\right] \cdot \frac{\langle k_1 \rangle - \langle k_2 \rangle}{2}. \tag{25}$$

Thus, we have

$$\hat{r}(r, c) = \frac{\left\langle \hat{k}_1 \right\rangle}{\left\langle \hat{k}_2 \right\rangle} = \frac{2r + (1 - r) \cdot (1 - \sqrt{c})^{2/r}}{2 - (1 - r) \cdot (1 - \sqrt{c})^{2/r}}. \tag{26}$$

Noticing that we can approximate $\hat{r}$ by $\hat{r} \approx r^c$, and approximate $\hat{v}$ by $\hat{v} \approx c \cdot v$ for simplicity in practice. Moreover, we also designed nonlinear approximation with a higher precision. However, the approximations that are more precise but more complex would only slightly improve the final results (see Supplementary Note 7 for the evaluations of the approximations).

Next, we explore the expression of $\mathcal{H}$ with parameters $c$, $r$, and $v$. Supposing that links between different nodes are independent, we obtain

$$\mathcal{H}(\mathbf{M}|A^{\mathcal{O}}) = \sum_{i=1}^{N} \sum_{j=1}^{N} \left[ \mathcal{H}(M_{ij}^1|A_{ij}^{\mathcal{O}}) + \mathcal{H}(M_{ij}^2|A_{ij}^{\mathcal{O}}) \right]. \tag{27}$$

For the OR-aggregation mechanism, $\mathcal{H}(M_{ij}^1|A_{ij}^{\mathcal{O}})$ and $\mathcal{H}(M_{ij}^2|A_{ij}^{\mathcal{O}})$ are both equal to 0 if $A_{ij}^{\mathcal{O}} = 0$. When $A_{ij}^{\mathcal{O}} = 1$, we have

$$E[\mathcal{H}(M_{ij}^\alpha|A_{ij}^{\mathcal{O}})] = -\bar{p}_\alpha \cdot \ln \bar{p}_\alpha - (1 - \bar{p}_\alpha) \cdot \ln(1 - \bar{p}_\alpha), \ \alpha = 1, 2, \tag{28}$$

where

$$\bar{p}_1 = E[P(M_{ij}^1 = 1|A_{ij}^{\mathcal{O}} = 1)] = \frac{\hat{v} + \hat{r}}{1 + \hat{r}}, \tag{29}$$

and

$$\bar{p}_2 = E[P(M_{ij}^2 = 1|A_{ij}^{\mathcal{O}} = 1)] = \frac{1 + \hat{v} \cdot \hat{r}}{1 + \hat{r}}. \tag{30}$$

Thus, we obtain

$$E[\mathcal{H}(\mathbf{M}|A^{\mathcal{O}})] = -|A^{\mathcal{O}}| \cdot \left[ \bar{p}_1 \cdot \ln \bar{p}_1 + (1 - \bar{p}_1) \cdot \ln(1 - \bar{p}_1) + \bar{p}_2 \cdot \ln \bar{p}_2 + (1 - \bar{p}_2) \cdot \ln(1 - \bar{p}_2) \right], \tag{31}$$

We empirically find that the accuracy of the reconstruction is linearly determined by the product of $\mathcal{H}$ and scaling factor $\rho$, i.e.,

$$\text{Accuracy} \approx 1 - \rho \cdot \mathcal{H}, \tag{32}$$

where

$$\rho = \frac{1}{2 \ln 2 \cdot |A^{\mathcal{O}}|} \cdot \left(1 - \frac{1 - v}{1 + v} \cdot c^s\right). \tag{33}$$

Notice that

$$E[P(M_{ij}^2 = 1|M_{ij}^1 = 1)] = \frac{v \cdot (1 + r)}{r \cdot (1 + v)}, \tag{34}$$

and

$$E[P(M_{ij}^1 = 1|M_{ij}^2 = 1)] = \frac{v \cdot (1 + r)}{1 + v}. \tag{35}$$

When observing an edge in layer $\alpha$, the probability that the edge exists in the other layer satisfies

$$1 - E[P(M_{ij}^\beta = 1|M_{ij}^\alpha = 1)] = 1 - \frac{\frac{v \cdot (1+r)}{r \cdot (1+v)} + \frac{v \cdot (1+r)}{1+v}/r}{1 + 1/r} = \frac{1 - v}{1 + v}, \ \alpha, \beta \in \{1, 2\}, \ \alpha \neq \beta. \tag{36}$$

Thus, the term $(1 - v)/(1 + v)$ in Eq. (33) indicates the fraction of uncertainty that can be reduced by partial observations, and $s$ describes the scale how partial observations can reduce the uncertainty of links in testing set.

**Budget allocation.** We consider the case when we have different fractions of partial observations in each layer denoted by $c_1$ and $c_2$. We denote $\mathcal{H}(M^1|A^{\mathcal{O}})$ by $\mathcal{H}_1$ for simplicity, and use $\bar{c} = (c_1 + c_2)/2$ to denote the given budget. Empirically, we can predict the accuracy by the function $F$ defined by

$$F(c_1, c_2) = 1 - \frac{1 - c_1}{1 - c_1 + (1 - c_2)/\hat{r}} \cdot \rho_1 \cdot \mathcal{H}_1 - \frac{(1 - c_2)/\hat{r}}{1 - c_1 + (1 - c_2)/\hat{r}} \cdot \rho_2 \cdot \mathcal{H}_2, \tag{37}$$

where

$$\rho_1 = \frac{1}{2 \ln 2 \cdot |A^{\mathcal{O}}|} \cdot \left(1 - \frac{1 - v}{1 + v} \cdot c_2^s\right), \tag{38}$$

and

$$\rho_2 = \frac{1}{2 \ln 2 \cdot |A^{\mathcal{O}}|} \cdot \left(1 - \frac{1 - v}{1 + v} \cdot c_1^s\right). \tag{39}$$

Once a certain budget ($\bar{c}$) is given, we regard the function $F$ as a unary function of $c_1$, i.e.,

$$F(c_1) = 1 - \frac{1 - c_1}{1 - c_1 + (1 - 2\bar{c} + c_1)/\hat{r}} \cdot \frac{1}{2 \ln 2 \cdot |A^{\mathcal{O}}|} \cdot \left[1 - \frac{1 - v}{1 + v} \cdot (2\bar{c} - c_1)^s\right] \cdot \mathcal{H}_1$$
$$- \frac{(1 - 2\bar{c} + c_1)/\hat{r}}{1 - c_1 + (1 - 2\bar{c} + c_1)/\hat{r}} \cdot \frac{1}{2 \ln 2 \cdot |A^{\mathcal{O}}|} \cdot \left(1 - \frac{1 - v}{1 + v} \cdot c_1^s\right) \cdot \mathcal{H}_2. \tag{40}$$

We next study the property of $F$ with $c_1$, and we will first prove that $\mathcal{H}_1 \geq \mathcal{H}_2$ here. According to the definition,

$$\mathcal{H}_\alpha = -\bar{p}_\alpha \cdot \ln \bar{p}_\alpha - (1 - \bar{p}_\alpha) \cdot \ln(1 - \bar{p}_\alpha), \ \alpha = 1, 2, \tag{41}$$

where

$$\bar{p}_1 = \frac{\hat{v} + \hat{r}}{1 + \hat{r}}, \ \bar{p}_2 = \frac{1 + \hat{v} \cdot \hat{r}}{1 + \hat{r}}. \tag{42}$$

Notice that the function

$$f(x) = -x \cdot \ln x - (1 - x) \cdot \ln(1 - x) \tag{43}$$

is a monotone increasing function when $0 < x \leq \frac{1}{2}$, and a monotone decreasing function when $\frac{1}{2} \leq x < 1$. For $\bar{p}_2$, we have

$$\bar{p}_2 = \frac{1 + \hat{v} \cdot \hat{r}}{1 + \hat{r}} \geq \frac{1}{1 + \hat{r}} \geq \frac{1}{2}. \tag{44}$$

When $\bar{p}_1 \leq \frac{1}{2}$,

$$\bar{p}_2 - (1 - \bar{p}_1) = \hat{v} \geq 0, \tag{45}$$

indicating $1/2 \leq 1 - \bar{p}_1 \leq \bar{p}_2$. Thus, $f(1 - \bar{p}_1) \geq f(\bar{p}_2)$, i.e., $\mathcal{H}_1 \geq \mathcal{H}_2$. When $\bar{p}_1 \geq \frac{1}{2}$,

$$\bar{p}_2 - \bar{p}_1 = \frac{(1 - \hat{v}) \cdot (1 - \hat{r})}{1 + \hat{r}} \geq 0, \tag{46}$$

indicating $1/2 \leq \bar{p}_1 \leq \bar{p}_2$. Thus, $f(\bar{p}_1) \geq f(\bar{p}_2)$, i.e., $\mathcal{H}_1 \geq \mathcal{H}_2$.

Then, we will consider the maxima of function $F$. When $\bar{c} \leq 1/2$ ($c_1 \in [0, 2\bar{c}]$), we have

$$F(0) = 1 - \frac{1}{2 \ln 2 \cdot |A^{\mathcal{O}}|} \cdot \left\{ \frac{\hat{r}}{\hat{r} + 1 - 2\bar{c}} \cdot \left[1 - \frac{1 - v}{1 + v} \cdot (2\bar{c})^s\right] \cdot \mathcal{H}_1 - \frac{1 - 2\bar{c}}{\hat{r} + 1 - 2\bar{c}} \cdot \mathcal{H}_2 \right\}, \tag{47}$$

and

$$F(2\bar{c}) = 1 - \frac{1}{2 \ln 2 \cdot |A^{\mathcal{O}}|} \cdot \left\{ \frac{\hat{r}(1 - 2\bar{c})}{\hat{r} + 1 - 2\hat{r}\bar{c}} \cdot \mathcal{H}_1 - \frac{1}{\hat{r} + 1 - 2\hat{r}\bar{c}} \cdot \left[1 - \frac{1 - v}{1 + v} \cdot (2\bar{c})^s\right] \cdot \mathcal{H}_2 \right\}. \tag{48}$$

Thus, we have

$$F(0) - F(2\bar{c}) = \frac{4\hat{r}\bar{c}(\bar{c} - 1) \cdot (\mathcal{H}_1 - \mathcal{H}_2) + \frac{1-v}{1+v} \cdot (2\bar{c})^s \cdot [\hat{r}(\hat{r} + 1 - 2\hat{r}\bar{c})\mathcal{H}_1 - (r + 1 - 2\bar{c})\mathcal{H}_2]}{2 \ln 2 \cdot |A^{\mathcal{O}}| \cdot (\hat{r} + 1 - 2\bar{c}) \cdot (\hat{r} + 1 - 2\hat{r}\bar{c})} \leq 0, \tag{49}$$

indicating $F(0) \leq F(2\bar{c})$.

When $\bar{c} > 1/2$ ($c_1 \in [2\bar{c} - 1, 1]$), we have

$$F(2\bar{c} - 1) = 1 - \frac{1}{2 \ln 2 \cdot |A^{\mathcal{O}}|} \cdot \frac{2v}{1 + v} \cdot \mathcal{H}_1, \tag{50}$$

and

$$F(1) = 1 - \frac{1}{2\ln 2 \cdot |A^{\mathcal{O}}|} \cdot \frac{2v}{1+v} \cdot \mathcal{H}_2. \tag{51}$$

Thus, we have

$$F(2\bar{c} - 1) - F(1) = \frac{1}{2\ln 2 \cdot N(N-1)} \cdot \frac{2v}{1+v} \cdot (\mathcal{H}_2 - \mathcal{H}_1) \le 0, \tag{52}$$

indicating $F(2\bar{c} - 1) \le F(1)$.

**Synthetic networks generation.** To have a deep exploration of the framework proposed in this article, we generate several synthetic networks with various network characteristics for performance evaluation. Here we mainly focus on two-layer networks with different $r$, $v$, and $\cos\langle \vec{d^1}, \vec{d^2} \rangle$ as we defined in the Results section.

As shown in Fig. 4b, we test synthetic networks ranging $0 < r \le 1$ and $0 \le v \le 1/2$. We first clarify how to generate a multiple network with a given $r^*$ and $v^*$. When given $r^*$ and $v^*$, we generate the adjacency matrix $M^1$ (the first layer in the multiplex network) by the Erdős–Rényi model, which indicates the edge $M_{ij}^1$ between any two nodes $i$ and $j$ submitted to the Bernoulli distribution

$$P(M_{ij}^1 = k) = \begin{cases} p, \text{if } k = 1 \\ 1 - p, \text{if } k = 0 \end{cases}. \tag{53}$$

Without loss of generality, we take $p = \frac{5}{N-1}$ such that the average degree $\langle k_1 \rangle = 5$ to avoid being too sparse. Then, we generate the adjacency matrix $M^2$ (the second layer in the multiplex network) by a specific way, where $M_{ij}^2$ is submitted to the distribution

$$P(M_{ij}^2 = k | M_{ij}^1 = 1) = \begin{cases} \frac{v^* \cdot (r^*+1)}{r^* \cdot (v^*+1)}, \text{if } k = 1 \\ 1 - \frac{v^* \cdot (r^*+1)}{r^* \cdot (v^*+1)}, \text{if } k = 0 \end{cases}, \forall i \ne j \tag{54}$$

and

$$P(M_{ij}^2 = k | M_{ij}^1 = 0) = \begin{cases} \frac{5}{N-6} \cdot \frac{1 - v^* \cdot r^*}{r^* \cdot (v^*+1)}, \text{if } k = 1 \\ 1 - \frac{5}{N-6} \cdot \frac{1 - v^* \cdot r^*}{r^* \cdot (v^*+1)}, \text{if } k = 0 \end{cases}, \forall i \ne j. \tag{55}$$

Next we prove that the parameters $r$ and $v$ of the generated multiplex network **M** satisfy $r = r^*$ and $v = v^*$.

Obviously, the expectation for average degree $\langle k_1 \rangle$ satisfies

$$E(\langle k_1 \rangle) = \frac{2}{N} \cdot \frac{N \cdot (N-1)}{2} \cdot \frac{5}{N-1} = 5, \tag{56}$$

and the expectation for average degree $\langle k_2 \rangle$ satisfies

$$E(\langle k_2 \rangle) = \frac{2}{N} \cdot \frac{N \cdot (N-1)}{2} \\ \cdot \left\{ \frac{5}{N-1} \cdot \frac{v^* \cdot (r^*+1)}{r^* \cdot (v^*+1)} + \left(1 - \frac{5}{N-1}\right) \cdot \frac{5}{N-6} \cdot \frac{1 - v^* \cdot r^*}{r^* \cdot (v^*+1)} \right\} = \frac{5}{r^*}. \tag{57}$$

Thus,

$$E(r) = \frac{E(\langle k_1 \rangle)}{E(\langle k_2 \rangle)} = r^*. \tag{58}$$

Then, the expectation of $|\mathcal{E}_1 \cap \mathcal{E}_2|$ satisfies

$$E(|\mathcal{E}_1 \cap \mathcal{E}_2|) = \frac{5N}{2} \cdot \frac{v^* \cdot (r^*+1)}{r^* \cdot (v^*+1)}, \tag{59}$$

and the expectation of $|\mathcal{E}_1 \cup \mathcal{E}_2|$ satisfies

$$E(|\mathcal{E}_1 \cup \mathcal{E}_2|) = \frac{5N}{2} + \left[ \frac{N \cdot (N-1)}{2} - \frac{5N}{2} \right] \cdot \frac{5}{N-6} \cdot \frac{1 - v^* \cdot r^*}{r^* \cdot (v^*+1)}, \tag{60}$$

Thus,

$$E(v) = \frac{E(|\mathcal{E}_1 \cap \mathcal{E}_2|)}{E(|\mathcal{E}_1 \cup \mathcal{E}_2|)} = v^*. \tag{61}$$

As shown in Fig. 4c, we test a number of synthetic multiplex networks ranging $0 < \cos\langle \vec{d^1}, \vec{d^2} \rangle < 1$. Since $\cos\langle \vec{d^1}, \vec{d^2} \rangle$ is determined by degree sequences of the two layers, we generate multiplex networks with given expectation of degree sequences. Specifically, we first randomly generate positive vectors $\vec{d^1}$ and $\vec{d^2}$ such that the inner product $\vec{d^1} \cdot \vec{d^2}$ ranges from 0 to 1. Then, we generate a number of multiplex networks with each layer being generated by the given degree sequence $\vec{d}^\alpha$. For adjacency matrix $M^\alpha$, the element $M_{ij}^\alpha$ is submitted to a Bernoulli distribution, i.e., $P(M_{ij}^\alpha = 1) = \frac{d^\alpha(i) \cdot d^\alpha(j)}{\| \vec{d}^\alpha \|_1 - 1}$. This generating process can also provide multiplex networks of different $0 < r_h \le 1$ as shown in the Supplementary Fig. 12a, b, c, since we can also generate the degree sequences with a given variance.

## Data availability

All data needed to evaluate the conclusions in the paper are available online as follows. The C. elegans multiplex connectome dataset used in this study is available at https://comunelab.fbk.eu/data.php. The London multiplex transportation network is available at https://comunelab.fbk.eu/data.php. The temporal social interactions at the SFHH (La Société française d'Hygiène Hospitalière) conference is available at www.sociopatterns.org/datasets/sfhh-conference-data-set/. The multiplex GPI (genetic and protein interactions) network of the Saccharomyces Pombe is available at https://comunelab.fbk.eu/data.php. The Yeast landscape multiplex interaction networks of genes is available at https://comunelab.fbk.eu/data.php. The multiplex air transportation network of Europe is available at http://complex.unizar.es/ātnmultiplex/. The multiplex air transportation network of the U.S.A. is available at http://stat-computing.org/dataexpo/2009/the-data.html. The temporal network of Wikipedia users editing each other's Talk page is available at http://snap.stanford.edu/data/wiki-talk-temporal.html. The CollegeMsg temporal social network is available at http://snap.stanford.edu/data/CollegeMsg.html.

## Code availability

Code reproducing the key results of this paper is available from the corresponding author upon reasonable request.

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

## Acknowledgements

We thank Prof. H. Eugene Stanley for enlightening discussions and constructive suggestions. We would like to additionally thank the anonymous reviewers for their comprehensive feedback. This work was supported by NSFC 62088101 Autonomous Intelligent Unmanned Systems, NSFC under grant No. U1909207, and Key Research and Development Program of Zhejiang Province (Grant No. 2021C03037). Shlomo H. thanks the Israel Science Foundation (Grant No. 189/19), the NSF-BSF (Grant No. 2019740), the EU H2020 project RISE (Project No. 821115), the EU H2020 DIT4TRAM, and DTRA (Grant No. HDTRA-1-19-1-0016) for financial support. J.G. acknowledges the support of the U.S. National Science Foundation under Grant No. 2047488 and the Rensselaer-IBM AI Research Collaboration.

## Author contributions

J.C., Shibo H., and J.G. conceived and designed the project. M.W. performed theoretical derivation and numerical simulation. J.C., Shibo H., J.G., and Shlomo H. contributed to the theoretical interpretation of the empirical data. All authors analyzed the results. M.W., J.C., Shibo H., and J.G. wrote the manuscript. Y.S. and Shlomo H. edited the manuscript.

## Competing interests

The authors declare no competing interests.
