## [Peer Review File · Communications Physics]

Reviewers' comments:

Reviewer #1 (Remarks to the Author):

Accurate reconstruction of multiplex networks is a classical problem that has attracted wide attention in network science. Fortunately, this work provides new perspective that can achieve optimal reconstruction only by some partial observations. It takes obvious contribution to this field. I have some comments as follows, wish to help improve this work.

Major comments.

1. The newly proposed framework appears effective that enables to achieve accurate reconstruction from partial observations, so how to certify its low-complexity with respect to your Q1? And how to determine the border of the partial observations?
2. Numerous real-world and artificial networks have been employed in this work, it seems that most of them are two layers, I suggest adding some multiplayer networks.
3. The authors have compared their results with three link prediction methods, this is necessary. Then I still have some doubts that if these methods are benchmark or advanced?

Minor comments.

1. The components in Figure 3 seem disordered.
2. The number under Figure 4 is wrongly located.
3. In Figure 4 (b), it seems the Pearson coefficient can be negative. And the color mark (r) fails to contain all the colors, such as red and pink.

Reviewer #2 (Remarks to the Author):

In their paper "Discrimination reveals reconstructability of multiplex networks from partial observations," Wu et al. propose a novel statistical approach to generate a multiplex network from partial observations and aggregate networks. The authors show the accuracy of their method and explore the robustness of reconstructability against various parameters of partial observations.

Overall, I find this extensive theoretical study interesting and timely in the context of generating and predicting multilayer properties of networks with uncertainties. I have several questions regarding the reported method, which do not reduce the quality of the manuscript.

1. Concerning the aggregate networks, it's not quite clear for me how the number of layers L of the hidden multiplex structure is defined? Is it based on some kind of deduction, prior knowledge about the networked system, or derivation using other mathematical methods?
2. In the subsection Framework for reconstructing multiplex layer structures, the authors state that the optimization process uses some network parameter θ . Further analyses exploit the node's degree k_i as such a parameter. Is the proposed method limited to the multiplex network reconstruction based on degree-sequences, or are the other parameters like centrality-based ones also suitable?
3. Optional remark. To gain readability, I recommend the authors replace bar plots with lines in Fig.3 d,e.
4. In the caption to Fig.3 f, should it be ' $q=0.5$ ' instead of ' $c=0.5$ '?

5. The text of the manuscript contains several grammatical errors or typos. Careful reconsideration is required.

Reviewer #3 (Remarks to the Author):

The manuscript introduces a method for reconstructing (inferring) layers of a multiplex network when given the aggregate network and partial observations of individual layers. The term partial observations means that all links are known among a fraction of a layer's nodes. The proposed method is based on likelihood maximization assuming a network model (e.g., the stochastic block model) and the authors show that a simple iterative maximization method can find the likelihood function's maximum. The reconstruction accuracy is found to tightly correlate with an entropy-based discrimination indicator. Optimal budget allocation and the behavior of three distinct dynamical processes on reconstructed networks are studied too.

The article seems quite solid to me (although there are some technical questions listed below) but I feel that the writing can be improved considerably as the clarity of what is being assumed and what is achieved is often lacking.

Major points:

1) The abstract is quite long but still not really clear. What does "partial observations" mean in the second sentence, for example? It would be useful if the assumed input data are clear from the abstract. Also, the allocation of observations is mentioned later on and one might guess what that might be in this context but it would have been really good to clearly explain what are the basic assumptions and what is the main task in the beginning.

2) The first paragraph of the results describes the notation and that is alright. The second paragraph starts with "The first step of the reconstruction is to find the most probable value of θ by maximizing the posterior probability $P(\theta|A^0, \Gamma)$, where θ is the network model parameter." but it is not clear which network parameter, and consequently which likelihood function, do you have in mind. The lacking clarifications here make it impossible to understand the actual content of equations (1) and (2) that thus remain abstract concepts. E.g., having some network observations, assuming some network model, and using model parameters $\theta_1, \dots, \theta_n$ (not a single parameter as you have), it is indeed possible to write down the probability distribution over all feasible multiplex structures M . How to do that remains unclear and the Methods section, albeit providing more equations, does not elucidate it. Only at the end of page 5, it is clarified that the proposed method is in fact model-specific and what are the parameters for various possible model choices. In my view, the abstract and pages 1--5 need to be considerably reorganized to make all assumptions and definitions clear.

3) With respect to the necessity to assume a model for network reconstruction, I am missing a discussion of whether the configuration model is adequate for all empirical networks. If not, how to recognize when not to use the configuration model and which other model to use instead?

4) On page 6, you write in passing "After estimating degree sequences D, \dots " but this is a point that deserves to be clarified to allow potential readers to reproduce your steps. How are those estimates obtained, what is their final form, and how well do they work in practice for different values of c , for

example? I see that the subsection "Mean-field approximation" has some related results but there are many different things done there, and some of the natural questions (such as how well the estimates work) are not addressed at all.

5) In relation to the subsection "Mean-field approximation", there are further things that require clarification. Already the newly introduced variables are confusing: r is "the ratio of the average degrees", for example. The same question pertains to the overlap of edges---the overlap between what? Finally, Equation (25) is approximated with r^c based on the agreement for $c = 0$ and $c = 1$ but it is not examined how good is the approximation between these two extreme values is, nor how much does it affect the results.

6) The authors do not use state of art information filtering metrics. The area under the precision-recall curve is superior to AUC for imbalanced datasets (networks tend to be sparse, so the imbalance problem is very real here) and the Matthews correlation coefficient is another popular choice (<https://www.ncbi.nlm.nih.gov/pmc/articles/PMC6941312>). It is unclear which of the presented results and conclusions remain valid after more appropriate metrics are used.

7) While the introduction and notation are written for a general number of layers, several important results later are formulated only for two layers and it is not clear how/if they would generalize to more layers [e.g., $s_{\{M\}}$ is found to correlate linearly with $\cos(d_1, d_2)$].

8) I am confused by the apparent accuracy of some empirically motivated results. I understand, for example, that one can find a linear empirical dependence between the accuracy and the information entropy. How is it possible the relatively complex analytical form of the proportionality factor, ρ ? Can we be sure that it is going to work well for all possible parameter values and network topologies? (I see that Fig. 4b compares the analytical formula and empirical accuracy results but that does not answer all questions mentioned above.)

Minor points:

1) The abstract opens with "The ultimate proof of an excellent method for predicting links in multiplex networks is reflected in its ability to reconstruct them accurately." - When you say "ultimate proof", I wonder what else than the ability to reconstruct a network makes an excellent method?

2) Some sentences are difficult to read or grammatically incorrect. A thorough text revision is necessary to make the paper's message really clear to the readers.

3) Sums over M in Eq. (3) (and elsewhere) should be over M that are consistent with A^0 and Γ , right? It could be useful to highlight this somehow.

4) In Figure 3f, you first say that $c = 0.5$ is fixed but then say that c varies (the latter is the case).

5) In the last section, Predicting dynamic processes in multiplex networks, the paper becomes quite terse and descriptive. For example, it is often stated that the results are worse in some parameter range but no attempts are made at explaining what is the reason for this, whether this described behavior in general (i.e., reconstructed networks become inadequate for a given process in various empirical networks).

6) In the Discussion, you say "to avoid the ergodic methods"---why should be ergodic methods avoided in this context?

7) In Figure 6, it could be useful to compare the results obtained with the newly proposed reconstruction method with the best-performing existing method (PRE 2019?).

8) The main contribution of your method is, as far as I got it, that it combines partially observed layers with a completely observed aggregate network for the network reconstruction problem. Is the manuscript title informative in this respect?

Article title: Discrimination reveals reconstructability of multiplex networks from partial observations

Manuscript Number: OMMSPHYS-21-0627

Response to the comments of Reviewer #1

Reviewer #1 (Remarks to the Author):

Accurate reconstruction of multiplex networks is a classical problem that has attracted wide attention in network science. Fortunately, this work provides new perspective that can achieve optimal reconstruction only by some partial observations. It takes obvious contribution to this field. I have some comments as follows, wish to help improve this work.

We thank the reviewer for the valuable comments. We have provided the following responses to address the reviewer's comments.

Major comments.

1.1 The newly proposed framework appears effective that enables to achieve accurate reconstruction from partial observations, so how to certify its low-complexity with respect to your Q1?

We thank the reviewer for pointing out these technical details that we might not clarify. According to our derivation in SI (Eq. 5), we obtained the distribution of multiplex structure

$$Q(M) = \frac{\mathbf{1}_{\{\varphi(M)=A^O\}} \cdot \mathbf{1}_{\{\Gamma \in M\}} \cdot \prod_{i<j} \prod_{\alpha} (p_{ij}^{\alpha})^{M_{ij}^{\alpha}} \cdot (1-p_{ij}^{\alpha})^{1-M_{ij}^{\alpha}}}{\sum_M \mathbf{1}_{\{\varphi(M)=A^O\}} \cdot \mathbf{1}_{\{\Gamma \in M\}} \cdot \prod_{i<j} \prod_{\alpha} (p_{ij}^{\alpha})^{M_{ij}^{\alpha}} \cdot (1-p_{ij}^{\alpha})^{1-M_{ij}^{\alpha}}},$$

where the term $\mathbf{1}_{\{\varphi(M)=A^O\}}$ and $\mathbf{1}_{\{\Gamma \in M\}}$ are indicative functions: $\mathbf{1}_{\{\cdot\}}$ equals to 1 if the condition in the brace is satisfied, and 0 otherwise. Thus, the term $\mathbf{1}_{\{\varphi(M)=A^O\}}$ describes the case where the multiplex structure M generates the aggregate network A^O , and the term $\mathbf{1}_{\{\Gamma \in M\}}$ indicates the case where M satisfies the partial observations Γ . The term $\prod_{i<j} \prod_{\alpha} (p_{ij}^{\alpha})^{M_{ij}^{\alpha}} \cdot (1-p_{ij}^{\alpha})^{1-M_{ij}^{\alpha}}$ indicates the likelihood of a given structure of M , where the edge probability between nodes i and j in layer α is denoted by p_{ij}^{α} . Notice that the denominator is the summation over all possible multiplex networks M that satisfy the constraints. The computational complexity $O(L \cdot m \cdot (2^L - 1)^m)$ is an exponential function of the number of observed edges m , since there are $(2^L - 1)^m$ possible items in the event space (Fig. 1a). Here, L is the number of layers. Assuming the independence of each link (and in each layer), we can extract the product operator, obtaining

$$Q(M) = \prod_{i<j} \frac{\mathbf{1}_{\{\varphi(M_{ij})=A_{ij}^O\}} \cdot \prod_{\alpha} \mathbf{1}_{\{\Gamma_{ij}^{\alpha} \in M_{ij}^{\alpha}\}} \cdot (p_{ij}^{\alpha})^{M_{ij}^{\alpha}} \cdot (1-p_{ij}^{\alpha})^{1-M_{ij}^{\alpha}}}{\sum_{M_{ij}} \mathbf{1}_{\{\varphi(M_{ij})=A_{ij}^O\}} \cdot \prod_{\alpha} \mathbf{1}_{\{\Gamma_{ij}^{\alpha} \in M_{ij}^{\alpha}\}} \cdot (p_{ij}^{\alpha})^{M_{ij}^{\alpha}} \cdot (1-p_{ij}^{\alpha})^{1-M_{ij}^{\alpha}}}.$$

Therefore, the probability space can be present as the product of m probability spaces, where there are totally $(2^L - 1)$ possible items in each event space (Fig. 1b), indicating that there are $m \cdot (2^L - 1)$ parameters. Since the computational complexity per iteration scales as $O(L \cdot m^2(2^L - 1))$ (L is a constant), the computational complexity drops from exponential to polynomial in the computation of $Q(M)$. Intuitively, we have separated the potential event space of A^0 into the product of event spaces of every link of A^0 , which drops the computational complexity $O(a^m)$ from exponential to a polynomial complexity $O(m^2)$ as we mentioned above.

Figure 1: A schematic illustration for the complexity dropping. (a) shows an example of the aggregate network consisting of three nodes (i,j,k) and two edges between them. For a two-layer multiplex network, there are 9 (i.e., 3^2) events in the event space, leading to an exponentially computational complexity, i.e., $O(3^m)$. (b) shows the independent assumption for the edge between i and j and the edge between j and k . In this case, the event space (i,j,k) can be presented as the product of two independent event spaces, each of which has 3 events. Therefore, the computational complexity has a linear growth, i.e., $O(3 \cdot m)$.

1.2 And how to determine the border of the partial observations?

As for the border of partial observations, we sample the partial observations by vertex sampling for a given fraction c of partial observation in the manuscript. For each value of c , we repeat the reconstruction 100 times to avoid randomness, and drawn error bars in all figures of the manuscript. Specifically, for a large undirected network, there are $\frac{N^2}{2}$ possible pairs of nodes, where N is the number of nodes. Note that, once we have observed these $\frac{N^2}{2}$ pairs, the network is certain and we do not need to infer the edge probability any more. Then, we randomly choose $n = \sqrt{c} \cdot N$ nodes, and the

subgraph consisting of these nodes includes $\frac{(\sqrt{c} \cdot N)^2}{2} = c \cdot \frac{N^2}{2}$ edges in the original network. We have elaborated this in the revised version.

2. Numerous real-world and artificial networks have been employed in this work, it seems that most of them are two layers, I suggest adding some multilayer networks.

Thanks for your suggestion. Our framework (introduced in Result I of the manuscript) works for multiplex networks with any number of layers. We took two layers as an illustration to analyze the intrinsic relationship between reconstruction accuracy and structure patterns. We have taken the reviewer's suggestion and added more results about real-world multiplex networks with more than two layers. For example, the European air transportation network has three layers (Ryanair, Lufthansa, EasyJet), and the air transportation network in the U.S. has four layers (SkyWest, Southwest, American Eagle, American Airlines). All the details of the network involved are listed in the Supplementary Information. In addition, analysis and experiments for more multilayer networks are left for future work.

3. The authors have compared their results with three link prediction methods, this is necessary. Then I still have some doubts that if these methods are benchmark or advanced?

Yes, these state-of-the-art works are advanced in the study of link prediction in multilayer networks. Moreover, these excellent works well on inference tasks by partial observations. Specifically, De Bacco et al. (PRE2017) have proposed a generative model and an efficient expectation-maximization algorithm, which allows performing inference tasks such as community detection and link prediction. It works for multiplex networks with groups, but it may fail in networks without group-based structures [1]. Tarres-Deulofeu et al. (PRE2019) introduced two stochastic block models for multilayer and temporal networks and developed scalable algorithms for inferring the parameters of these models, bringing a more accurate (in terms of standard metrics such as precision and recall) result than those of single-layer models [2]. However, our reconstruction method is different from the link prediction method, since the studies above did not integrate the information of the aggregated network. Thus, the state-of-the-art works were considered as baselines, which provides valuable insights into how aggregated topology information helps reconstruction.

Minor comments.

4. The components in Figure 3 seem disordered.

We have modified the disordered components in Figure 3 in the revision.

5. The number under Figure 4 is wrongly located.

The layout problem led to some numbers being wrongly located, which we have modified.

6. *In Figure 4 (b), it seems the Pearson coefficient can be negative. And the color mark (r) fails to contain all the colors, such as red and pink.*

We are sorry that some marks in Figure 4(b) are out of bounds, which may confuse readers that the Pearson coefficient can be negative. We have modified the figures in the revision. In addition, the color mark (r) displays the results for synthetic networks, while the red and pink marks display the results for specific real-world networks.

Response to the comments of Reviewer #2

Reviewer #2 (Remarks to the Author):

In their paper "Discrimination reveals reconstructability of multiplex networks from partial observations," Wu et al. propose a novel statistical approach to generate a multiplex network from partial observations and aggregate networks. The authors show the accuracy of their method and explore the robustness of reconstructability against various parameters of partial observations. Overall, I find this extensive theoretical study interesting and timely in the context of generating and predicting multilayer properties of networks with uncertainties. I have several questions regarding the reported method, which do not reduce the quality of the manuscript.

We thank the reviewer for the valuable comments. We have provided the following responses to address the reviewer's comments.

1. *Concerning the aggregate networks, it's not quite clear for me how the number of layers L of the hidden multiplex structure is defined? Is it based on some kind of deduction, prior knowledge about the networked system, or derivation using other mathematical methods?*

Thanks for your comments. In this work, we focus on the reconstruction of multiplex structure and assume that we have known the number of layers. Thus, predicting the number of layers is not our main goal in this article. In fact, several recent works have made an attempt to answer the question of whether an observed single-layer network is an outcome of a multilayer process (e.g., an aggregate by OR/AND mechanism), and further inferred the number of hidden layers [3,4]. Specifically, once they observed the aggregate network A^0 , they can calculate the posterior probability $P(A^0|L)$. By performing the maximum likelihood estimation, they can identify the most probable number of layers L for the multiplex network. Therefore, we can adopt these approaches to calculate the number of layers and then include them in our reconstruction, when the number of layers L of the hidden multiplex structure is unknown.

2. *In the subsection Framework for reconstructing multiplex layer structures, the authors state that the optimization process uses some network parameter θ . Further analyses exploit the node's degree k_i as such a parameter. Is the proposed method limited to the multiplex network reconstruction based on degree-sequences, or are the other parameters like centrality-based ones also suitable?*

Thanks for pointing out this. The proposed method does not limit to the multiplex network reconstruction based on degree-sequences, since the proposed framework works for multiplex networks with any form of network parameters. Therefore, other parameters like centrality-based ones are also suitable, as long as there is a mechanism to generate a network by these parameters. Since it is not possible to enumerate all generative models, we did not include the results from other generative models. We adopted the configurational model to provide a clear and concrete understanding of our reconstruction framework since it is a relatively general model that allows an arbitrary degree sequence. Moreover, though our analysis was illustrated by the configuration model that was based on the degree sequence, the empirical results did not rely on the degree sequence. Practically, we have tested a mass of synthetic networks with various degree sequences in our simulations, and we find a stable relationship between reconstruction accuracy and multiplex structure.

3. *Optional remark. To gain readability, I recommend the authors replace bar plots with lines in Fig.3 d,e.*

We have modified the figures as suggested in the revision.

4. *In the caption to Fig.3 f, should it be ' $q=0.5$ ' instead of ' $c=0.5$ '?*

Yes, we are sorry to have made this mistake and have checked the manuscript carefully in the revision.

5. *The text of the manuscript contains several grammatical errors or typos. Careful reconsideration is required.*

We thank the reviewer for carefully reading the first part of the manuscript. Though the main results are right, we are sorry for the poor presentation and typos that confused the reviewer. We have significantly improved the presentation of the article and corrected several typos. We believe the manuscript is now clear to understand.

Response to the comments of Reviewer #3

Reviewer #3 (Remarks to the Author):

The manuscript introduces a method for reconstructing (inferring) layers of a multiplex network when given the aggregate network and partial observations of individual

layers. The term *partial observations* means that all links are known among a fraction of a layer's nodes. The proposed method is based on likelihood maximization assuming a network model (e.g., the stochastic block model) and the authors show that a simple iterative maximization method can find the likelihood function's maximum. The reconstruction accuracy is found to tightly correlate with an entropy-based discrimination indicator. Optimal budget allocation and the behavior of three distinct dynamical processes on reconstructed networks are studied too. The article seems quite solid to me (although there are some technical questions listed below) but I feel that the writing can be improved considerably as the clarity of what is being assumed and what is achieved is often lacking.

We thank the reviewer for the comments. We have significantly improved the manuscript to address all issues and concerns raised by the reviewer and provided the following responses to address the reviewer's comments.

Major points:

1. *The abstract is quite long but still not really clear. What does "partial observations" mean in the second sentence, for example? It would be useful if the assumed input data are clear from the abstract. Also, the allocation of observations is mentioned later on and one might guess what that might be in this context but it would have been really good to clearly explain what are the basic assumptions and what is the main task in the beginning.*

We thank the reviewer for pointing out the unclear presentations in the abstract. In fact, the "partial observations" here referred to the set of observed links, such as a subgraph in a network. We strongly agree with the reviewer that it is good to clearly explain what are the basic assumptions and what is the main task in the beginning. Thus, we have modified our representations in the revised abstract, such that the assumed input data and the main tasks are clear for the general audience. For example, our basic assumptions are the aggregate topology and the set of observed links, and the main task is to predict unobserved links in the multiplex network.

- 2.1 *The first paragraph of the results describes the notation and that is alright. The second paragraph starts with "The first step of the reconstruction is to find the most probable value of θ by maximizing the posterior probability $P(\theta|A^0, \Gamma)$, where θ is the network model parameter." but it is not clear which network parameter, and consequently which likelihood function, do you have in mind.*

We are sorry that we have not clarified several important concepts in the manuscript, which might confuse readers. With respect to network model parameter, in single-layer networks, it is hypothesized that a network is generated by some process such that the probability of generating a network with adjacency matrix A is $P(A|\theta)$, where θ represents the parameters of such a process [5]. Thus, θ is a set of all the parameters

in the network model. For example, the Erdos-Renyi random network model generates a network with a single parameter θ , which describes the link probability between any pair of nodes. For another example, the stochastic block model can generate networks with a group structure, where the set of network parameters include, K (the number of groups), δ_i (the probability that node belongs to a group i), W_{ij} (the link probabilities between groups i and j , where $i \neq j$). Thus, the network model parameter θ is ig dimensional when describing the proposed general method. Once the network model is determined in practice, network model parameter θ refers to specific forms. In our experiments, we adopt the configuration model, and the parameter θ in the model refers to the degree sequences D in the multiplex network, and the corresponding likelihood function refers to $P(A^O, \Gamma|D)$.

2.2 The lacking clarifications here make it impossible to understand the actual content of equations (1) and (2) that thus remain abstract concepts. E.g., having some network observations, assuming some network model, and using model parameters $\theta_1, \dots, \theta_n$ (not a single parameter as you have), it is indeed possible to write down the probability distribution over all feasible multiplex structures M . How to do that remains unclear and the Methods section, albeit providing more equations, does not elucidate it. Only at the end of page 5, it is clarified that the proposed method is in fact model-specific and what are the parameters for various possible model choices. In my view, the abstract and pages 1--5 need to be considerably reorganized to make all assumptions and definitions clear.

In the manuscript, we only reserved the general forms of the equations to avoid complicated notations. Thus, we put the specific forms with derivations in Supplementary Note II. For example, the explicit form of the probability distribution over all feasible multiplex structures M is

$$Q(M) = P(M|A^O, \Gamma, \theta) = \frac{\mathbf{1}_{\{\varphi(M)=A^O\}} \mathbf{1}_{\{\Gamma \in M\}} \prod_{i < j} \prod_{\alpha} (p_{ij}^{\alpha})^{M_{ij}^{\alpha}} (1 - p_{ij}^{\alpha})^{1 - M_{ij}^{\alpha}}}{\sum_M \mathbf{1}_{\{\varphi(M)=A^O\}} \mathbf{1}_{\{\Gamma \in M\}} \prod_{i < j} \prod_{\alpha} (p_{ij}^{\alpha})^{M_{ij}^{\alpha}} (1 - p_{ij}^{\alpha})^{1 - M_{ij}^{\alpha}}}$$

where the term $\mathbf{1}_{\{\varphi(M)=A^O\}}$ and $\mathbf{1}_{\{\Gamma \in M\}}$ are indicative functions. The indicative function $\mathbf{1}_{\{\cdot\}}$ equals to 1 if the condition in brace is satisfied, and 0 otherwise. Thus, the term $\mathbf{1}_{\{\varphi(M)=A^O\}}$ describes the case where the multiplex structure M generates the aggregate network, and the term $\mathbf{1}_{\{\Gamma \in M\}}$ indicates the case where M satisfies the partial observations Γ . The term $\prod_{i < j} \prod_{\alpha} (p_{ij}^{\alpha})^{M_{ij}^{\alpha}} \cdot (1 - p_{ij}^{\alpha})^{1 - M_{ij}^{\alpha}}$ indicates the likelihood of a given structure of M , where the edge probability between nodes i and j in layer α is denoted by p_{ij}^{α} . Obviously, the denominator is the summation over all possible multiplex networks M that satisfy the constraints. Notice that the term p_{ij}^{α}

describes the link probability between nodes i and j in layer α , and we employ the configuration model for an illustration in this article, i.e.,

$$p_{ij}^\alpha = \frac{d^\alpha(i) \cdot d^\alpha(j)}{\|\bar{d}^\alpha\|_1 - 1}.$$

We strongly agree with the reviewer that it is necessary to clarify all mentioned notations and abstract concepts. As the reviewer suggested, we have reorganized the abstract and relevant results to make all assumptions and definitions clear.

3. With respect to the necessity to assume a model for network reconstruction, I am missing a discussion of whether the configuration model is adequate for all empirical networks. If not, how to recognize when not to use the configuration model and which other model to use instead?

In the manuscript, we provided the general framework and took the configuration model as an example to provide a clear and concrete understanding of our reconstruction framework. The configuration model is a relatively general model that allows an arbitrary degree sequence. When we have no prior knowledge about the generating mechanism of the network, the configuration model is a good choice. Further, we emphasize that any specific analytical model can be included in our proposed framework. In practice, we need prior knowledge about the network to recognize which model to use. For example, if a network exhibits group structure, we may hypothesize that the network is generated from a stochastic block model [6], where the parameters of this model tell us about community structure.

4.1 On page 6, you write in passing "After estimating degree sequences D, \dots " but this is a point that deserves to be clarified to allow potential readers to reproduce your steps. How are those estimates obtained, what is their final form, and how well do they work in practice for different values of c , for example?

Thanks for your comments. We have presented more results in this revision and due to space limitation, we put detailed derivations in the Supplementary Information. We provide detailed explanations in the following.

Specifically, we consider a set consisting of all possible multiplex network structures M that can generate the aggregate topology A^O and partial observations Γ . We calculate the posterior probability $P(M|A^O, \Gamma, \theta)$, where θ is the parameter of the generative model (e.g., degree sequence D in configuration model). Thus, the explicit form of $P(M|A^O, \Gamma, D^*)$ is

$$P(M|A^O, \Gamma, D^*) = \frac{\mathbf{1}_{\{\varphi(M)=A^O\}} \cdot \mathbf{1}_{\{\Gamma \in M\}} \cdot \prod_{i < j} \prod_{\alpha} (p_{ij}^\alpha)^{M_{ij}^\alpha} \cdot (1-p_{ij}^\alpha)^{1-M_{ij}^\alpha}}{\sum_M \mathbf{1}_{\{\varphi(M)=A^O\}} \cdot \mathbf{1}_{\{\Gamma \in M\}} \cdot \prod_{i < j} \prod_{\alpha} (p_{ij}^\alpha)^{M_{ij}^\alpha} \cdot (1-p_{ij}^\alpha)^{1-M_{ij}^\alpha}},$$

which is also denoted by $Q(M)$, and is shown in the Eq. 5 SI. Thus, we take partial derivative of the above equation with respect to each element in D and find the value of them at which the partial derivative equals to zero, i.e.,

$$\frac{\nabla}{\nabla d^\alpha} \sum_M Q(M) \cdot \ln \prod_{i < j} \left(\frac{d^\alpha(i) \cdot d^\alpha(j)}{2m-1} \right)^{M_{ij}^\alpha} \cdot \left(1 - \frac{d^\alpha(i) \cdot d^\alpha(j)}{2m-1} \right)^{1-M_{ij}^\alpha} = 0, \alpha = 1, 2, \dots, L,$$

where ∇ is the differential operator. For a large network, the number of edges m is sufficiently large, and can be regarded as a constant. Then, we obtain

$$\sum_j d^\alpha(i) \cdot d^\alpha(j) = \sum_j (2m-1) \cdot Q(M_{ij}^\alpha) \cdot M_{ij}^\alpha, \forall i, \alpha.$$

and

$$d^\alpha(i) = \frac{2m-1}{2m-d^\alpha(i)} \sum_j Q(M_{ij}^\alpha) \cdot M_{ij}^\alpha, \forall i, \alpha.$$

As we have assumed that m is sufficiently large, we obtain the final form of the degree sequence

$$d^\alpha(i) = \sum_j Q(M_{ij}^\alpha) \cdot M_{ij}^\alpha = \sum_j E(M_{ij}^\alpha), \forall i, \alpha.$$

As for the question of how the estimation works in practice for different values of c , we proved that the variance of the estimator designed in our framework decreases as the fraction of partial observations c increases, and it reaches the CRLB (Cramer-Rao lower bound) when the network size N approaches infinity, indicating an asymptotic efficient estimation from the theoretic point of view.

4.2 I see that the subsection "Mean-field approximation" has some related results but there are many different things done there, and some of the natural questions (such as how well the estimates work) are not addressed at all.

In the subsection "Mean-field approximation", we made several approximations. In general, mean-field theory studies the behavior of high-dimensional random (stochastic) models by studying a simpler model that approximates the original by averaging over degrees of freedom. Thus, we tried to explore an explicit relationship between reconstruction accuracy and network structure by the mean-field approximation. For example, the information entropy of a multiplex network M by aggregate observation A^O is defined by

$$\mathcal{H}(M|A^O) = \sum_{i=1}^N \sum_{j=1}^N [\mathcal{H}(M_{ij}^1|A_{ij}^O) + \mathcal{H}(M_{ij}^2|A_{ij}^O)],$$

which has a very high degrees of freedom. When employing the mean-field approximation, we are averaging over degrees of freedom, and approximating $P(M_{ij}^1 = 1|A_{ij}^O = 1), \forall i, j$, by $E[P(M_{ij}^1 = 1|A_{ij}^O = 1)]$. After these approximations, we obtained high Pearson correlations between theoretical prediction and empirical results, indicating they work very well.

5.1 In relation to the subsection "Mean-field approximation", there are further things that require clarification. Already the newly introduced variables are confusing: r is "the ratio of the average degrees", for example. The same question pertains to the overlap of edges---the overlap between what?

Thanks for pointing out this. The relevant variables are defined in the subsection "Reconstructability determined by the discrimination indicator". Specifically, r is the ratio of average degrees of two layers, i.e., $r = \frac{\langle k_1 \rangle}{\langle k_2 \rangle}$, where $\langle k_1 \rangle$ and $\langle k_2 \rangle$ are the average degrees in layer 1 and layer 2, respectively. Further, v is the overlap of the edge sets in two layers of a multiplex network, which is defined by the Jaccard index of E_1 and E_2 that indicate the two edge sets in the two layers, i.e., $v = |E_1 \cap E_2| / |E_1 \cup E_2|$. We have taken the reviewer's suggestion and emphasized these important definitions in different subsections.

5.2 Finally, Equation (25) is approximated with r^c based on the agreement for $c = 0$ and $c = 1$ but it is not examined how good is the approximation between these two extreme values is, nor how much does it affect the results.

We thank the reviewer for pointing these approximations that we might miss to clarify. In the original article, we claimed that we approximate $\hat{r}(r, c)$ by $\hat{r} \approx r^c$, since $\hat{r}(r, 0) = 1$ and $\hat{r}(r, 1) = r$, and we approximate $\hat{v}(v, c)$ by $\hat{v} \approx c \cdot v$, since $\hat{v}(v, 0) = 0$ and $\hat{v}(v, 1) = v$. However, we made this approximation not only by these two extreme values, but also by other evidence. For example, we have examined the approximations between these two extreme values on nine empirical networks in Figure 2 (also see in Supplementary Figure 7). The curves show the approximations between these two extreme values in different networks, and the Pearson coefficients shown in Figure 3 are calculated to examine how good is the approximations made here. Specifically, the Pearson coefficient between \hat{r} and r^c reaches 0.89 in all empirical networks, and the Pearson coefficient between \hat{v} and $c \cdot v$ reaches 0.77 in all empirical networks.

Figure 2: *The mesoscale structure revealed in reconstructed multiplex networks. The average degree, and the overlap of edges in the reconstructed network for nine real-world networks with c increasing.*

Figure 3: The performance of the mean-field approximation. (a) we approximate $\hat{r}(r, c)$ by $\hat{r} \approx r^c$, and the Pearson coefficient between them reaches 0.89 in all empirical networks. (b) we approximate $\hat{v}(v, c)$ by $\hat{v} \approx c \cdot v$, and the Pearson coefficient between them reaches 0.77 in all empirical networks.

6. The authors do not use state of art information filtering metrics. The area under the precision-recall curve is superior to AUC for imbalanced datasets (networks tend to be sparse, so the imbalance problem is very real here) and the Matthews correlation coefficient is another popular choice (<https://www.ncbi.nlm.nih.gov/pmc/articles/PMC6941312>). It is unclear which of the presented results and conclusions remain valid after more appropriate metrics are used.

We thank the reviewer for the state of art metrics. We have complemented the empirical results for these necessary simulations in the revision, which are shown in Figure 4. The AUPRC (area under the precision-recall curve) performed well in the two empirical networks. Notice that the MCC (Matthews correlation coefficient) ranges in the interval $[-1, +1]$, with extreme values -1 and $+1$ reached in case of perfect misclassification and perfect classification, respectively, while $MCC=0$ is the expected value for the coin tossing classifier. The results of MCC are consistent with the other metrics considered.

Figure 4: The other two metrics are used to measure the reconstruction performance. The mentioned AUPRC and Matthews correlation coefficient with accuracy, precision, recall and AUC are tested against c (percentage of partial observations). These results are obtained by repeating 1,00 times from two real-world networks: (a) *C. elegans* connectome; (b) London transportation network;

7. While the introduction and notation are written for a general number of layers, several important results later are formulated only for two layers and it is not clear how/if they would generalize to more layers [e.g., $s_{\{M\}}$ is found to correlate linearly with $\cos(d_1, d_2)$].

As the reviewer mentioned, our framework (introduced in Result I of the manuscript) works for multiplex networks with any number of layers. Further, we took multiplex networks with two layers as an illustration and made an attempt to analyze the intrinsic relationship between reconstruction accuracy and structure patterns. At this time, however, several important results so far cannot be generalized to more layers directly, such as the cosine similarity mentioned, since it remains a challenging issue to analyze structure patterns in a general multiplex network with any number of layers. We strongly agree with the reviewer that it is of great significance to generalize our theoretical analysis to more layers, which are left for future work. We have made this clear in the revision.

8.1 I am confused by the apparent accuracy of some empirically motivated results. I understand, for example, that one can find a linear empirical dependence between the accuracy and the information entropy. How is it possible the relatively complex analytical form of the proportionality factor, ρ ?

Thanks for your comments. We have found the relationship between the reconstruction accuracy and the information entropy empirically. Then, we have performed further analysis on these forms. For example, the proportionality factor ρ has a relatively complex analytical form of

$$\rho = \frac{1}{2 \ln 2 \cdot |A^O|} \cdot \left(1 - \frac{1-v}{1+v} c^s\right).$$

Notice that the term $\frac{1}{2 \ln 2 \cdot |A^O|}$ is a normalization constant. For any one observed edge in the aggregate network of a multiplex network with two layers, the maximum information entropy of the potential structure space is $2 * \left(-\frac{1}{2} \cdot \ln \frac{1}{2} - \frac{1}{2} \cdot \ln \frac{1}{2}\right) = 2 \ln 2$. Further, once we observe $|A^O|$ edges in the aggregate network A^O , the maximum information entropy of the potential structure space becomes $2 \ln 2 \cdot |A^O|$. Moreover, notice that

$$E[P(M_{ij}^2 = 1 | M_{ij}^1 = 1)] = \frac{v \cdot (1+r)}{r \cdot (1+v)},$$

and

$$E[P(M_{ij}^1 = 1 | M_{ij}^2 = 1)] = \frac{v \cdot (1+r)}{1+v}.$$

Thus, when observing an edge in one layer α , the probability that the edge exists in the other layer β satisfies

$$1 - E\left[P\left(M_{ij}^\beta = 1 \mid M_{ij}^\alpha = 1\right)\right] = 1 - \frac{\frac{v \cdot (1+r)}{r \cdot (1+v)} + \frac{v \cdot (1+r)}{1+v} / r}{1 + 1/r} = \frac{1-v}{1+v}.$$

Therefore, the term $\frac{1-v}{1+v}$ in the analytical form of ρ indicates the fraction of uncertainty that can be reduced by partial observations, and s describes the scale how partial observations can reduce the uncertainty of links in testing set. In summary, the reconstructability is determined by the discrimination indicator $(1 - \rho \cdot \mathcal{H})$ from both microscale and mesoscale views, indicating that the reconstruction can be predicted accurately either ρ or \mathcal{H} is small.

8.2 Can we be sure that it is going to work well for all possible parameter values and network topologies? (I see that Fig. 4b compares the analytical formula and empirical accuracy results but that does not answer all questions mentioned above.)

We obtained our analytical results by theoretical derivation, where we obtained these conclusions by several necessary assumptions: a) The network is sufficient large; b) Each link in the network is independent. Thus, it is going to work well for all possible parameter values and network topologies if the assumptions hold. However, if a network did not match these assumptions, it might be not going to work well. For example, we have assumed that the network size is sufficiently large, and the independence of each link in each layer. Therefore, our analyze may fail in some small-scale networks and networks with a high degree correlation. We have made this clear in the revision.

Minor points:

9. *The abstract opens with "The ultimate proof of an excellent method for predicting links in multiplex networks is reflected in its ability to reconstruct them accurately." - When you say "ultimate proof", I wonder what else than the ability to reconstruct a network makes an excellent method?*

Thanks for your comments. As this sentence may confuse readers, we have revised it to: *An excellent method for predicting links in multiplex networks is reflected in its ability to reconstruct them accurately.*

10. *Some sentences are difficult to read or grammatically incorrect. A thorough text revision is necessary to make the paper's message really clear to the readers.*

We are sorry for the poor presentations that make several sentences difficult to read. We have significantly improved the presentation of the article and corrected several typos.

11. *Sums over M in Eq. (3) (and elsewhere) should be over M that are consistent with $A \neq 0$ and Γ , right? It could be useful to highlight this somehow.*

Yes, thanks for pointing out this. We have highlighted this in the revision.

12. *In Figure 3f, you first say that $c = 0.5$ is fixed but then say that c varies (the latter is the case).*

We are sorry to make this typo that confused the reviewer. In fact, it should be ' $q = 0.5$ ' instead of ' $c = 0.5$ '.

13. *In the last section, predicting dynamic processes in multiplex networks, the paper becomes quite terse and descriptive. For example, it is often stated that the results are worse in some parameter range but no attempts are made at explaining what is the reason for this, whether this described behavior in general (i.e., reconstructed networks become inadequate for a given process in various empirical networks).*

We thank the reviewer for the suggestion. We have modified in the revision and provided detailed description about this.

14. *In the Discussion, you say "to avoid the ergodic methods"---why should be ergodic methods avoided in this context?*

We thank the reviewer for pointing out these technical details that we did not clarify. According to our derivation in SI (Eq. 5), we obtained the distribution of multiplex structure

$$Q(M) = \frac{\mathbf{1}_{\{\varphi(M)=A^O\}} \cdot \mathbf{1}_{\{\Gamma \in M\}} \cdot \prod_{i < j} \prod_{\alpha} (p_{ij}^{\alpha})^{M_{ij}^{\alpha}} \cdot (1 - p_{ij}^{\alpha})^{1 - M_{ij}^{\alpha}}}{\sum_M \mathbf{1}_{\{\varphi(M)=A^O\}} \cdot \mathbf{1}_{\{\Gamma \in M\}} \cdot \prod_{i < j} \prod_{\alpha} (p_{ij}^{\alpha})^{M_{ij}^{\alpha}} \cdot (1 - p_{ij}^{\alpha})^{1 - M_{ij}^{\alpha}}},$$

where the term $\mathbf{1}_{\{\varphi(M)=A^O\}}$ and $\mathbf{1}_{\{\Gamma \in M\}}$ are indicative functions: $\mathbf{1}_{\{\cdot\}}$ equals to 1, if the condition in the brace is satisfied, and 0 otherwise. Thus, the term $\mathbf{1}_{\{\varphi(M)=A^O\}}$ describes if the multiplex structure M generates the aggregate network A^O , and the term $\mathbf{1}_{\{\Gamma \in M\}}$ indicates if M satisfies the partial observations Γ . The term $\prod_{i < j} \prod_{\alpha} (p_{ij}^{\alpha})^{M_{ij}^{\alpha}} \cdot (1 - p_{ij}^{\alpha})^{1 - M_{ij}^{\alpha}}$ indicates the likelihood of a given structure of M , where the edge probability between nodes i and j in layer α is denoted by p_{ij}^{α} . Notice that the denominator is the summation over all possible multiplex networks M that satisfy the constraints. The computational complexity $O(L \cdot m \cdot (2^L - 1)^m)$ is an exponential function of the number of observed edges m , since there are $(2^L - 1)^m$ possible items in the event space (L is the number of layers). Thus, it is impossible to calculate the distribution by ergodic methods. That is why it should be avoided.

15. In Figure 6, it could be useful to compare the results obtained with the newly proposed reconstruction method with the best-performing existing method (PRE 2019?).

Thanks for the suggestion. In the mentioned figure, we applied our method to predict several dynamics. In the mentioned figure, we tried to study the reconstruction performance with different scale of partial observation rates, and we did not provide full-scale analysis in this section then. However, we agree with the reviewer that it could be useful to compare the results obtained with the newly proposed reconstruction method with the best-performing existing method, which we have complemented the results in the revised supplementary information, which are also shown in Figure 5 here.

Figure 5: The performance of dynamic prediction with different methods. (a) The percolation processes of the reconstructed multiplex network ($c=0.5$) for *C. elegans* multiplex connectome. The x-axis denotes the occupied probability p and the y-axis denotes the size of GMCC when nodes are randomly removed with probability $1 - p$ in one layer. (b) A random walk process taking place on the reconstructed multiplex network and real multiplex network for London transportation network. The x-axis

denotes time t and the y -axis denotes coverage (the fraction of nodes that have been visited before a certain time) of n walkers starting from a set of random chosen nodes. (c) The spreading process on the reconstructed temporal network and real temporal network for the social interactions at the SFHH. The x -axis denotes the infection rate and the y -axis denotes the infected fraction.

16. The main contribution of your method is, as far as I got it, that it combines partially observed layers with a completely observed aggregate network for the network reconstruction problem. Is the manuscript title informative in this respect?

Generally, in the case of network reconstruction problems, a completely aggregate network is always observed. For example, several recent excellent works have made attempts to reconstruct the layer structure by an observed aggregate network, and attracted much attention [3,4]. Thus, we did not emphasize the aggregate observation but the partial observations in the manuscript title.

References

- [1] De Bacco C, Power E A, Larremore D B, et al. Community detection, link prediction, and layer interdependence in multilayer networks[J]. *Physical Review E*, 2017, 95(4): 042317.
- [2] Tarrés-Deulofeu M, Godoy-Lorite A, Guimerà R, et al. Tensorial and bipartite block models for link prediction in layered networks and temporal networks[J]. *Physical Review E*, 2019, 99(3): 032307.
- [3] Valles-Catala T, Massucci F A, Guimera R, et al. Multilayer stochastic block models reveal the multilayer structure of complex networks[J]. *Physical Review X*, 2016, 6(1): 011036.
- [4] Lacasa L, Mariño I P, Míguez J, et al. Multiplex decomposition of non-Markovian dynamics and the hidden layer reconstruction problem[J]. *Physical Review X*, 2018, 8(3): 031038.
- [5] Newman M E J. Estimating network structure from unreliable measurements[J]. *Physical Review E*, 2018, 98(6): 062321.
- [6] Holland P W, Laskey K B, Leinhardt S. Stochastic blockmodels: First steps[J]. *Social networks*, 1983, 5(2): 109-137.

Reviewers' comments:

Reviewer #1 (Remarks to the Author):

The authors have answered all my questions. I think the paper is acceptable.

Reviewer #2 (Remarks to the Author):

I am satisfied with the authors' response and the revised manuscript. The authors have fully addressed all the issues raised. Thus I recommend the paper for publishing in its current form if accepted by the other referees.

Reviewer #3 (Remarks to the Author):

I wish to thank the authors for their extensive reply to my points. I find the response to be satisfactory and the revised manuscript more clear. There are only two points that have caught my attention:

1) In Figure 3 on page 12 of the response letter, you provide empirical evidence for the approximations $\hat{r} \approx r^c$ and $\hat{v} \approx c*v$. I think that the correlation values provide an overly simplified picture here: (a) The left panel has quite a high correlation value and the agreement is indeed mostly good but there is one network (Air Transp. US) where the curve does not agree with the prediction at all. So the approximation is not universally valid and there is something about that network that makes it fail. (b) Here the correlation value itself is not meaningful as it is influenced, I believe, by individual networks' curves ending at various points of the x-axis: London transp. with the lowest \hat{v} ends first, then US Air transp. with a higher \hat{v} , etc. I do not know what determines the endpoint (largest $c*v$) of each individual curve but it certainly greatly contributes to the presented overall correlation value. When looking at individual curves instead, a very different picture emerges where \hat{r} is either totally independent of $c*v$ (Social interact and Sacchpomb GPI) or \hat{v} grows with $c*v$ only in the very beginning (the first two data points) and then it remains flat.

In summary, the empirical evidence shows that the approximation for \hat{r} works in all networks but one and the approximation for \hat{v} does not work anywhere. Based on the actual importance of these results to the paper as a whole (that you can judge better than me), it would be good if they are commented on accurately and their implications are discussed.

2) Figure 5 on page 16 of the response letter compares the performance of the newly proposed method with other methods in terms of dynamical processes on the reconstructed network. According to the legend, your method (Reconstruction) performs considerably worse than two other methods (PRE2017 and PRE2019). Isn't the legend wrong here (as a comparison with Figure 6 in the manuscript could suggest)? If not, then this is bad news for your method and it is an issue that should be discussed and analyzed in the manuscript (instead of only showing this figure in Supplementary Information).

Article title: Discrimination reveals reconstructability of multiplex networks from partial observations

Manuscript Number: COMMSPHYS-21-0627

Response to the comments of Reviewer #1

Reviewer #1 (Remarks to the Author):

The authors have answered all my questions. I think the paper is acceptable.

We thank the reviewer for the acceptance.

Response to the comments of Reviewer #2

Reviewer #2 (Remarks to the Author):

I am satisfied with the authors' response and the revised manuscript. The authors have fully addressed all the issues raised. Thus I recommend the paper for publishing in its current form if accepted by the other referees.

We thank the reviewer for the acceptance.

Response to the comments of Reviewer #3

Reviewer #3 (Remarks to the Author):

I wish to thank the authors for their extensive reply to my points. I find the response to be satisfactory and the revised manuscript more clear. There are only two points that have caught my attention:

Thank the reviewer for pointing out these technical questions. We have provided the following responses to address them.

*1.1 In Figure 3 on page 12 of the response letter, you provide empirical evidence for the approximations $\hat{r} \approx r^c$ and $\hat{v} \approx c^*v$. I think that the correlation values provide an overly simplified picture here.*

Thanks for the comments. The correlation values in the previous response letter provide a simplified picture. Therefore, we offered more detailed analyses in the revision. We first present the approximated ratio of average degree \hat{r} of two layers ranging c from 0 to 1 for nine real-world networks, where the fraction of partial observations c indicates the ratio between the number of observed edges in layers and that of all edges in the multiplex network. The nine curves show different patterns, and there is no universal approximation for all empirical networks (at least for the nine networks we consider) (Figure 1a). The manuscript takes the approximation $\hat{r} \approx r^c$ for all considered networks in empirical calculations (Figure 1b), since it is not possible to design a specific approximation for each network. The Pearson coefficient between them reaches 0.89, indicating the approximation is valid for common empirical networks to a certain degree.

Figure 1: The performance of the approximation for \hat{r} . (a) The empirical results for the approximated ratio of average degree \hat{r} ranging c from 0 to 1 for nine real-world networks. (b) We approximated \hat{r} by $\hat{r} \approx r^c$ for all considered networks, and the Pearson coefficient between them reaches 0.89 in all empirical networks.

1.2 The left panel has quite a high correlation value and the agreement is indeed mostly good but there is one network (Air Transp. US) where the curve does not agree with the prediction at all. So the approximation is not universally valid and there is something about that network that makes it fail.

Thanks for the comments. Finding an approximation of the parameter is one of the intermediate steps to obtain the main result, which is the relationship between reconstruction accuracy and the discrimination indicator. Although the approximation is not universally valid, it does not affect our main result and final conclusions. Further, even though we have an approximation with higher estimation precision, it may not contribute to the analysis for the reconstructability distinctly. For example, as the reviewer mentioned, in one network (Air Transp. US), the curve does not agree with the prediction. In this revision, we performed a detail analysis for this individual network (Figure 2a), where the simplified approximation $\hat{r} \approx r^c$ reaches 0.79 in this network (Figure 2b). Further, we designed a new approximation with higher precision of \hat{r} by $\hat{r} \approx 1 + \left(r \cdot c^{\frac{1}{4}} - 1\right) \cdot c^{\frac{1}{4}}$ for this specific network. By employed the new approximation, we have a higher correlation with a Pearson coefficient of 0.98 (Figure 2c).

Figure 2: The performance of the two different approximations for \hat{r} in the air transportation multiplex network (U.S.). (a) The star markers indicate the empirical results for \hat{r} in the considered network. The pink curve indicates the approximation for $\hat{r}(r, c)$ by $\hat{r} \approx r^c$, and the blue curve indicates the approximation for $\hat{r}(r, c)$ by $\hat{r} \approx 1 + (r \cdot c^{\frac{1}{4}} - 1) \cdot c^{\frac{1}{4}}$. (b) We approximate $\hat{r}(r, c)$ by $\hat{r} \approx r^c$ for the considered network, and the Pearson coefficient between them reaches 0.79. (c) We approximate $\hat{r}(r, c)$ by $\hat{r} \approx 1 + (r \cdot c^{\frac{1}{4}} - 1) \cdot c^{\frac{1}{4}}$ for the considered network, and the Pearson coefficient between them reaches 0.98.

However, we did not recommend the new approximation ($\hat{r} \approx 1 + (r \cdot c^{\frac{1}{4}} - 1) \cdot c^{\frac{1}{4}}$), although it performs well on the estimation of \hat{r} . There are two reasons. First, the approximation with higher precision will not reach a better performance in the final results, i.e., the relationship between accuracy of reconstruction and the discrimination indicator $\rho \cdot \mathcal{H}$ for these two different approximations are the same, and the Pearson correlations are both 0.97 (Figure 3). Second, the approximation with higher precision has a precise estimation only in the air transportation multiplex network (U.S.), and it is difficult to be applied to other cases in practice because of the complex form.

Figure 3: The performance of the two different approximations for \hat{r} in the air transportation multiplex network (U.S.). The two different approximations show the

Pearson correlation between accuracy of reconstruction and the discrimination indicator $\rho \cdot \mathcal{H}$ are both 0.97.

1.3 Here the correlation value itself is not meaningful as it is influenced, I believe, by individual networks' curves ending at various points of the x-axis: London transp. with the lowest \hat{v} ends first, then US Air transp. with a higher \hat{v} , etc. I do not know what determines the endpoint (largest $c \cdot v$) of each individual curve but it certainly greatly contributes to the presented overall correlation value.

Thanks for your comments. Each point of an individual curve was determined by the coordinate $(c \cdot v, \hat{v})$, where \hat{v} is the approximation of v obtained from empirical experiments, and $c \cdot v$ is the proposed approximation. Thus, the endpoint of each individual curve indicates the coordinate $(c \cdot v, \hat{v})$ corresponding to the largest $c = 1$, i.e., (v, v) , when the network is totally observed. Thus, the endpoint of each individual curve is fixed on the line $y = x$, and it is true that these endpoints contribute to the correlation value because of the certainty. In a similar way for the approximation of \hat{v} , we make a simplified approximation, indicating the linear approximation $\hat{v} \approx c \cdot v$ for a convenient calculation in practice. However, the correlation value remains meaningful as it shows the approximation effect for overall considered empirical networks to a certain degree.

1.4 When looking at individual curves instead, a very different picture emerges where \hat{v} is either totally independent of $c \cdot v$ (Social interact and Sacchpomb GPI) or \hat{v} grows with $c \cdot v$ only in the very beginning (the first two data points) and then it remains flat.

For each individual curve, it is true that \hat{v} is either totally independent of $c \cdot v$ or \hat{v} grows with $c \cdot v$ only in the very beginning, since it is a linear approximation. This is because in this article the approximation focuses on finding a simple uniform scheme for overall empirical networks. Moreover, we take the reviewer's suggestion, and design a nonlinear approximation with a higher precision in this revision. As a matter of fact, the relationship between accuracy of reconstruction and the discrimination indicator increases slightly, which is analyzed in the revised Supplementary Information.

In our present results, specifically, we made a simplified approximation for \hat{v} , indicating the linear approximation $\hat{v} \approx c \cdot v$ in practice. This approximation reaches a Pearson correlation of 0.77 for the nine considered networks (*Figure 4a*), and the final results show the linear relationship between accuracy of reconstruction and the discrimination indicator $\rho \cdot \mathcal{H}$, where the Pearson correlation reaches 0.96 (*Figure 4b*).

Figure 4: The performance of the approximation for $\hat{v} \approx c \cdot v$. (a) We approximate $\hat{v}(v, c)$ by $\hat{v} \approx c \cdot v$, and the Pearson coefficient between them reaches 0.77 in all empirical networks. (b) The relationship between accuracy of reconstruction and the discrimination indicator $\rho \cdot \mathcal{H}$ for nine real-world networks is displayed, where the Pearson correlation is 0.96.

As mentioned above, we also made an approximation that is more precise for \hat{v} by $\hat{v} \approx c \cdot v^{\frac{1}{10}}$, reaching a higher correlation for all considered networks (Pearson coefficient 0.94) (Figure 5a). However, this nonlinear approximation only made a slightly improvement with the final result, increasing the Pearson correlation between accuracy of reconstruction and the discrimination indicator from 0.96 to 0.97 (Figure 5b). We put the results in the Supplementary Information.

Figure 5: The performance of the approximation for $\hat{v} \approx c \cdot v^{\frac{1}{10}}$. (a) We approximate $\hat{v}(v, c)$ by $\hat{v} \approx c \cdot v^{\frac{1}{10}}$, and the Pearson coefficient between them reaches 0.94 in all empirical networks. (b) The relationship between accuracy of reconstruction and the discrimination indicator $\rho \cdot \mathcal{H}$ for nine real-world networks is displayed, where the Pearson correlation is 0.97.

1.5 In summary, the empirical evidence shows that the approximation for \hat{r} works in all networks but one and the approximation for \hat{v} does not work anywhere. Based on the actual importance of these results to the paper as a whole (that you can judge better than me), it would be good if they are commented on accurately and their implications are discussed.

Thanks for the suggestions. As we analyzed above, although several empirical networks may not agree with the prediction with the simplified approximation, it provided a convenient way for calculation in practice. Moreover, approximations that are more precise but more complex would only slightly improve the results. Therefore, we could conclude that although several simplified approximations are made in empirical calculations, our main results and conclusions are untouched. The detailed analyses and their implications are complemented in the revised Supplementary Information.

2) Figure 5 on page 16 of the response letter compares the performance of the newly proposed method with other methods in terms of dynamical processes on the reconstructed network. According to the legend, your method (Reconstruction) performs considerably worse than two other methods (PRE2017 and PRE2019). Isn't the legend wrong here (as a comparison with Figure 6 in the manuscript could suggest)? If not, then this is bad news for your method and it is an issue that should be discussed and analyzed in the manuscript (instead of only showing this figure in Supplementary Information).

Thanks for pointing out this. We are sorry that we have made wrong legends in the mentioned figure in the response. We have corrected the legends in the figure here, where we found that the proposed method (Reconstruction) performs considerably better than two other methods (PRE2017 and PRE2019).

Figure 6: The performance of dynamic prediction with different methods. (a) The percolation processes of the reconstructed multiplex network ($c = 0.5$) for *C. elegans* multiplex connectome. The x-axis denotes the occupied probability p and the y-axis denotes the size of GMCC when nodes are randomly removed with probability $1 - p$ in one layer. **(b)** A random walk process taking place on the reconstructed multiplex network and real multiplex network for London transportation network. The x-axis denotes time t and the y-axis denotes coverage (the fraction of nodes that have been visited before a certain time) of n walkers starting from a set of random chosen nodes. **(c)** The spreading process on the reconstructed temporal network and real temporal network for the social interactions at the SFHH. The x-axis denotes the infection rate and the y-axis denotes the infected fraction.

REVIEWERS' COMMENTS:

Reviewer #3 (Remarks to the Author):

I would like to thank the authors for the answer and the modifications to the manuscript. I believe that clarifying the accuracy and impact of the approximations is a useful addition to the manuscript.